# WS$_2$ moiré superlattices derived from mechanical flexibility for hydrogen evolution reaction

Lingbin Xie[1], Longlu Wang[2✉], Weiwei Zhao[1], Shujuan Liu[1], Wei Huang ⬤ [1,3✉] & Qiang Zhao[1,2✉]

The discovery of moiré superlattices (MSLs) opened an era in the research of 'twistronics'. Engineering MSLs and realizing unique emergent properties are key challenges. Herein, we demonstrate an effective synthetic strategy to fabricate MSLs based on mechanical flexibility of WS$_2$ nanobelts by a facile one-step hydrothermal method. Unlike previous MSLs typically created through stacking monolayers together with complicated method, WS$_2$ MSLs reported here could be obtained directly during synthesis of nanobelts driven by the mechanical instability. Emergent properties are found including superior conductivity, special super-aerophobicity and superhydrophilicity, and strongly enhanced electro-catalytic activity when we apply 'twistronics' to the field of catalytic hydrogen production. Theoretical calculations show that such excellent catalytic performance could be attributed to a closer to thermo-neutral hydrogen adsorption free energy value of twisted bilayers active sites. Our findings provide an exciting opportunity to design advanced WS$_2$ catalysts through moiré superlattice engineering based on mechanical flexibility.

[1] State Key Laboratory of Organic Electronics and Information Displays & Jiangsu Key Laboratory for Biosensors, Institute of Advanced Materials (IAM) & Institute of Flexible Electronics (Future Technology), Nanjing University of Posts & Telecommunications, Nanjing, China. [2] College of Electronic and Optical Engineering & College of Microelectronics, Jiangsu Province Engineering Research Center for Fabrication and Application of Special Optical Fiber Materials and Devices, Nanjing University of Posts & Telecommunications (NUPT), Nanjing, China. [3] Frontiers Science Center for Flexible Electronics (FSCFE), MIIT Key Laboratory of Flexible Electronics (KLoFE), Northwestern Polytechnical University, Xi'an, Shaanxi, China. ✉email: wanglonglu@njupt.edu.cn; provost@nwpu.edu.cn; iamqzhao@njupt.edu.cn

Moiré superlattices (MSLs) created by a twist inspire a hot area of 'twistronics' for advanced materials sciences. With the help of periodic moiré patterns, MSLs could optimize the structure and energy band[1,2], resulting in many phenomena, including moiré phonon[3], moiré exciton[4–7], magnetism[8], topological edge states[9,10], unconventional superconductivity[11–15], Mott insulation[16–20] and so on. MSLs exhibit promising applications in electronics[21,22], optoelectronics[23], valleytronics[24], photonic[25,26], spintronics[27], and electrocatalysis[28–31].

The commonly used physical and chemical methods to fabricate MSLs through stacking together are quite complicated and require the use of specific substrate and experimental conditions[11–15,31–33]. Consequently, it is desirable to develop an easy and versatile strategy to construct MSLs and present an ideal model system for investigating the emergent properties. Recently, Eli Sutter's et al.[34] have developed a method for the preparation of van der Waals chiral nanowires distorted by layered crystals, extending the path of interlayer distortion to achieve MSLs from two-dimensional planes to one-dimensional nanowires. However, this method still needs the use of substrates, which are difficult to produce on a large scale. It is very meaningful to explore a class of van der Waals one-dimensional (1D) nanostructures of layered crystals, in which MSLs evolve naturally during synthesis without substrate. TMDs nanobelts, combining both the flexibility and unidirectional properties of 1D nanomaterials would enable the production of MSLs easily through spontaneous deformation. Herein, we successfully synthesize large-scale homogeneous MSLs based on the mechanical flexibility of $WS_2$ by a facile and reproducible one-pot hydrothermal method. MSLs could be well introduced to $WS_2$ along with their controllable growth. Ultrathin $WS_2$ nanobelts with high flexibility can spontaneously bend and twist into the helix nanocones arising from mechanical instability. The bending and twisting could cause the S–W–S layer to slip for the production of MSLs.

Furthermore, we find the emergent properties of nanocone-like $WS_2$ MSLs, such as superior conductivity, special super-aerophobicity and superhydrophilicity, which brought unexpected catalytic hydrogen production performance by comparison with various other $WS_2$ based electro-catalysts. The as-synthesized $WS_2$ MSLs electrocatalysts display an overpotential of 60 mV at a current density of 10 mA cm$^{-2}$ and a Tafel slope of 40 mV dec$^{-1}$. Meanwhile, the unique nanostructures of $WS_2$ MSLs with the superhydrophilic property for the rapid access of the electrolyte and the underwater super-aerophobic property further facilitate the fast mass transfer characteristics of $WS_2$ MSLs. The experimental results are supported by theoretical calculations and the underlying mechanism is ascribed to much more appropriate $\Delta G_H$ of twisted bilayers $WS_2$ active sites compared with that of normal bilayers $WS_2$.

## Results

**Synthesis and structural characterization of $WS_2$ MSLs.** Owing to the excellent mechanical properties of ultrathin 1D and 2D materials, various specific topological structures such as ripples, bends, scrolls, helixes, wrinkles, folds, and curls are shown in Supplementary Fig. 1, could spontaneously form by the thermodynamic and mechanical factors during the synthesis process. These unique topological structures may bring rich and excellent electronic properties. To achieve this goal, herein, a unique $WS_2$ topology deformed from nanobelts has been designed by a facile hydrothermal method.

Field-emission scanning electron microscope image (Fig. 1a) illustrated the uniformity of the as-prepared $WS_2$ MSLs at a large-scale view, consisting of numerous conical nanoarray with an average width of ~200 nm. The electrodes consist of 3D $WS_2$

MSLs with open space were in favor of electrolyte ion transport. The SEM-energy dispersive spectrometer element mapping images (Supplementary Fig. 2) showed the uniform coverage of W and S elements on the surface of the $WS_2$ nanoarray. As shown in Fig. 1b, the single conical tube was curled from nanobelts as indicated by scanning transmission electron microscopy. One end remains nanobelt, meanwhile, the other end has transformed into coin-like. In the synthesis process, the $WS_2$ nanobelts easily deform under the various unbalanced external forces and then twist into nanocones. Finite-element calculations of strain in a nanocone (Fig. 1c) showed the relative stress distribution with negligible strain at no twisted end and big strain at another largely twisted end. The strain introduced by twisting may contribute to activating the basal plane of the nanobelts by changing the electronic structure of catalytic active sites and facilitating mass transfer[35]. As shown in Fig. 1d, S–W–S layer slipping could be triggered by the mechanical instability, accompanying the generation of MSLs.

Moiré superlattice is created through stacking two monolayers together rotated with respect to each other, along wavelength periodic modulation results[36]. The van der Waals force of multi-layer $WS_2$ nanobelt is further weakened by the strain that is induced by the mechanical instability. $WS_2$ nanobelt would transform to a $WS_2$ nano-cone with layer slipping that could induce the formation of moiré patterns. The significant honeycomb-structured moiré patterns are found throughout the measured HRTEM images region in Fig. 2a, b. Figure 2c illustrated the simulated diagram of the $WS_2$ MSL atomic structure with a twisted $\theta$ (14°), which has high consistency with the structure shown in Fig. 2b. The corresponding FFT pattern of the HRTEM image in the sheet contains 12 {1100} spots, which constitute two hexagons (double sets of sixfold symmetry diffraction spots), as illustrated in Fig. 2d. The FFT patterns and IFFT images of HRTEM lattice images were performed, as shown in Fig. 2e, f, respectively. On the basis of the splitting spots in the FFT patterns (Fig. 2g), the moiré patterns in this region exhibited twist angles of 13.82°. We collected ten HRTEM images and their corresponding FFT images taken from different $WS_2$ nanocones to ensure the twist angle in Supplementary Fig. 3. All the values of the twist angles are in the range of 13°–14°. We build a model with the rotational stacking faults of 13.2° as shown in Supplementary Fig. 4, which is the calculable model closest to the rotational stacking faults of 13°–14° obtained by the experiment. The high consistency between the experimental HRTEM images and the simulated HRTEM images is presented in Supplementary Fig. 5. Dislocations of atomic planes and strain distributions of corresponding lattice planes were indicated by Geometric phase analysis images in Fig. 2h, i obtained from HRTEM image (Fig. 2a). It is obviously illustrated that the strain was introduced successfully by topology engineering based on mechanical flexibility.

2H (trigonal prismatic), 1T (octahedral) and 1T' (clustered W) phase were different phases of $WS_2$, and their atomic structure models were shown in Supplementary Fig. 6. Bending and twisting could induce the glide of S atoms in basal planes of $WS_2$ MSLs to generate a 1 T/1T' phase. X-ray photoelectron spectroscopy (XPS) and Raman spectroscopy were used to distinguish the 1T/1T' and 2H phase of $WS_2$. To make a better comparison, 1T'-$WS_2$ NSs were also successfully prepared and well-characterized by TEM, XRD, and XPS in Supplementary Figs. 7–9. The XPS characterization of 1T'-$WS_2$ NSs in Supplementary Fig. 9 showed that 1T' content in 1T'-$WS_2$ NSs reaches nearly 100%. As shown in Supplementary Fig. 10, two peaks at around 34.7 and 32.7 eV were characteristic of the 2H-$WS_2$ features corresponding to W4$f_{5/2}$ and W4$f_{7/2}$, respectively[37–39]. The new peaks of the 1T' phase clearly shifted toward lower binding

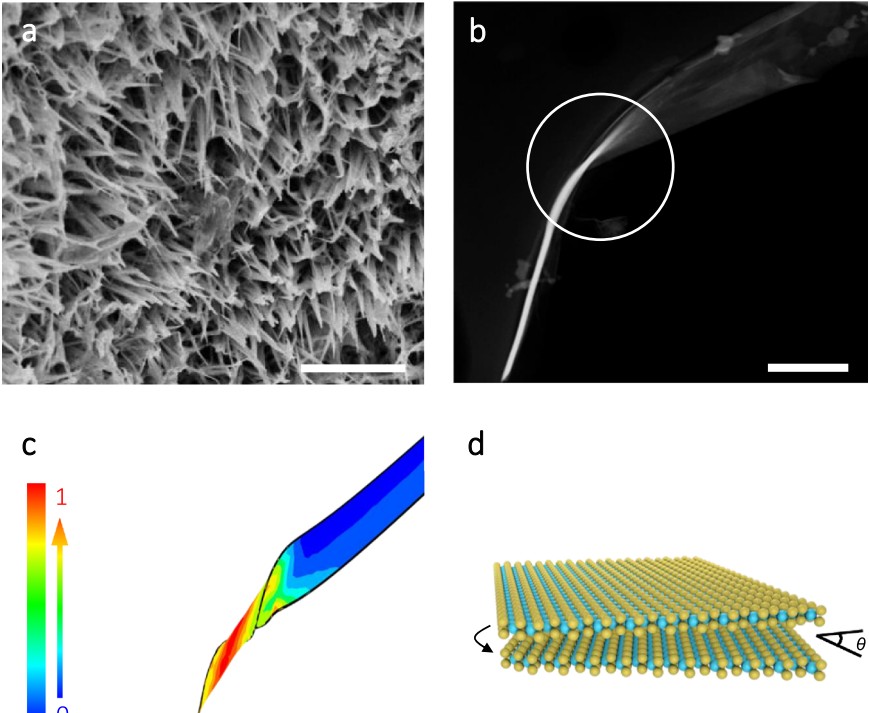

**Fig. 1 Morphological and structural characterizations. a** FESEM image of the as-prepared WS$_2$ nanoarrays. Scale bar, 2 μm. **b** STEM image of single screwed WS$_2$ nanobelt. Scale bar, 300 nm. **c** Finite-element calculations of strain in a nanocone. The color bar shows the relative scale of the strain distribution. **d** Schematic diagram of moiré superlattices formed by S–W–S layer slipping. Yellow and cyan balls represent S and W atoms, respectively.

energies (33.1 and 31.1 eV, corresponding to W4$f_{5/2}$ and W4$f_{7/2}$ of 1T'-WS$_2$ components). The result reveals the obvious formation of the 2H and metallic 1T' phase in WS$_2$ MSLs. Raman characterizations of all samples are shown in Supplementary Fig. 11. Two prominent peaks corresponding to the in-plane $E^1_{2g}$ and out-of-plane $A_{1g}$ modes of 2H-WS$_2$ are observed in WS$_2$ MSLs. The WS$_2$ MSLs sample also exhibits small peaks in the lower frequency region that correspond to the active modes of distorted 1T'-WS$_2$ NSs which are not allowed in the 2H-WS$_2$ NSs. The three peaks should be ascribed to the $J_1$–$J_3$ vibration modes of S–W–S bonds in 1T'-WS$_2$ phase, clearly demonstrating the coexistence of 1T' and 2H phases in WS$_2$ MSLs[38–41]. In addition, the HRTEM analysis unambiguously confirms the formation of 1T'@2H-WS$_2$ heterostructures (Supplementary Fig. 12). The 1T'-WS$_2$ structure could be locked by a collective elastic-deformation barrier from distortion against the transformation into the more stable 2H polymorph.

**Catalytic activity**. The as-prepared 2H-WS$_2$ NSs, 1T'-WS$_2$ NSs, WS$_2$ MSLs, and commercial Pt/C (20 wt%) were used to investigate the HER electrocatalytic performance. Polarization curves of these samples with a scan rate of 10 mV s$^{-1}$ in Ar-bubbled 0.5 M H$_2$SO$_4$ are shown in Fig. 3a and Supplementary Fig. 13. All electrochemical performance tests of catalysts were carried out on carbon fiber cloth (CFC). As shown in Supplementary Fig. 14, the bare CFC exhibits negligible electrocatalytic performance. As shown in Fig. 3a, the low overpotential of just 60 mV vs. RHE under the current density ($J = 10$ mA cm$^{-2}$) was needed for the WS$_2$ MSLs, which is smaller than those of other WS$_2$ samples, such as the as-prepared 2H-WS$_2$ NSs (248 mV vs. RHE) and 1T'-WS$_2$ NSs (212 mV vs. RHE), except for the commercial Pt/C (20 wt%). The Tafel slope suggests that the HER reaction of WS$_2$ MSLs may follow a similar Volmer–Heyrovsky mechanism and is closely related to electrochemical desorption[42–45], unlike the Pt/C

electrocatalyst (30 mV dec$^{-1}$) via the Volmer–Tafel mechanism (Fig. 3b). The as-prepared WS$_2$ MSLs exhibited much more excellent HER performance (e.g., low overpotential at $J = 10$ mA cm$^{-2}$ and small Tafel slope) than the reported representative non-precious HER electrocatalysts. (Supplementary Fig. 15 and Supplementary Tables 1 and 2).

To verify the long-term stability, the as-prepared WS$_2$ MSLs were tested in a prolonged run by chronoamperometry test (Fig. 3c). Compared to the stability of 1T'-WS$_2$ NSs (Supplementary Fig. 16), WS$_2$ MSLs have not weakened during the chronoamperometric response for 20 h.

Accurate determination of the electrochemically active surface area (ECSA) is vitally important to evaluate the electrocatalytic activity of catalyst. In general, the ECSA of each catalyst is estimated from measurements of the double-layer capacitance ($C_{dl}$). The $C_{dl}$ value of given electrocatalysts is determined by Cyclic voltammetry (CV) in a non-faradaic region or by electrochemical impedance spectroscopy (EIS)[46–48]. As shown in Supplementary Figs. 17–20, using the CV method, plotting the cathodic and anodic current as a function of the scan rate revealed a linear function, where the slope indicated the $C_{dl}$. As shown in Supplementary Fig. 21 and Supplementary Table 4, the fitting parameters of EIS further verified the $C_{dl}$ values of measured from the scan rate-dependent CVs (within 15% difference[46]). We assume the general specific capacitance of 60 μF cm$^{-2}$ to estimate ECSA from the $C_{dl}$ values of catalysts[49–51] (Detailed calculation and analysis can be found in Supplementary Note 1). The ECSA value of WS$_2$ MSLs (396.6 cm$^2$ ECSA) was much higher than that of 1T'-WS$_2$ NSs (253.3 cm$^2$ ECSA) and 2H-WS$_2$ NSs (190.0 cm$^2$ ECSA), indicating that the WS$_2$ MSLs possessed more enrichment of active sites for electrochemical hydrogen evolution (Fig. 3d and Supplementary Table 3).

Moreover, the fitted $R_{ct}$ values for WS$_2$ MSLs, 1T'-WS$_2$ NSs, and 2H-WS$_2$ NSs are 1.6, 3.4, and 11.2 Ω, respectively (Supplementary Fig. 22). The result suggests that the surface of

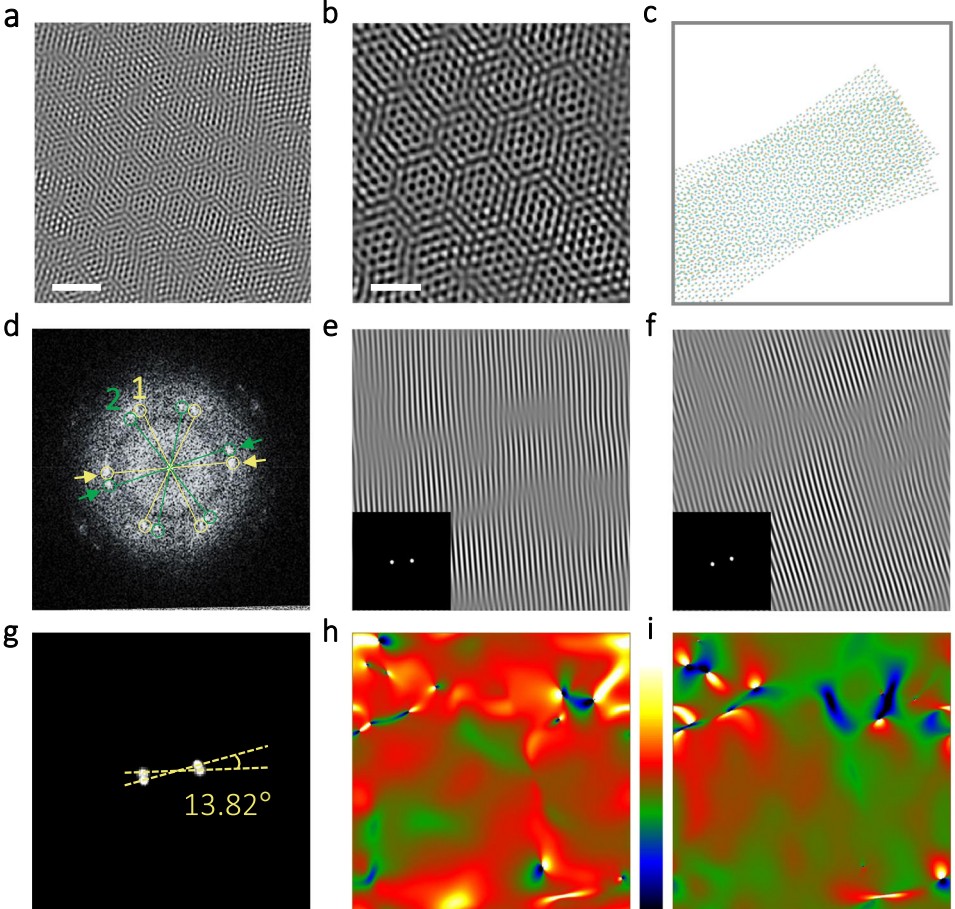

**Fig. 2 WS$_2$ MSLs induced by rotational stacking fault. a** High-resolution transmission electron microscopy (HRTEM) characterization of as-prepared WS$_2$ nanocone. Scale bar, 2 nm. **b** Enlarged HRTEM characterization. Scale bar, 1 nm. **c** Schematic diagram of the WS$_2$ MSLs. Meanwhile, a twist angle of 14° is set in the bilayer regions and distinctive moiré patterns are shown clear. Yellow and cyan balls represent S and W atoms. **d** The corresponding fast Fourier transform (FFT) pattern of (**b**) where the 12 spots constitute two hexagons that are marked with 1 and 2. Hexagon 1 is marked in yellow and hexagon 2 is in green. **e**, **f** Corresponding inverse FFT (IFFT) patterns of the FFT spots pointed by the arrows in (**d**). **g** The filtered FFT patterns corresponding to the IFFT patterns of (**e**) and (**f**). **h**, **i** Strain distributions of $e_{xx}$ and $e_{xy}$, respectively. (The color from green to dark blue and the color from red to bright yellow represent the compressive strain and tensile strain, respectively).

WS$_2$ MSLs has excellent interfacial charge transfer kinetics for electrocatalysis. To demonstrate the superior conductivity of WS$_2$ MSLs, the total potentials of WS$_2$ MSLs with misorientation angles of 13.2° were investigated by density functional theory (DFT) calculations. As shown in Supplementary Fig. 23, apparently, the potential barriers of WS$_2$ MSLs with different phases reduced in contrast with normally stacked bilayer WS$_2$, indicating that electron orbitals coupling in WS$_2$ MSLs became much stronger[28]. Thus, electrons transfer much more easily between two adjacent layers that would have good effect on HER catalytic properties of WS$_2$ MSL.

Exchange current density ($j_0$) was used to evaluate the HER activity of different WS$_2$ catalysts[52] (Supplementary Fig. 24 and Supplementary Table 3). The $j_0$ of 2.13 μA cm$^{-2}$ ECSA for the WS$_2$ MSLs sample surpasses the values of 1.55 μA cm$^{-2}$ ECSA for 1 T'-WS$_2$ NSs sample and 1.09 μA cm$^{-2}$ ECSA for 2H-WS$_2$ NSs sample, highlighting the electrochemical activity of WS$_2$ MSLs. The turnover frequency (TOF) was used to determine the intrinsic activity of WS$_2$ MSLs[49,50,53,54]. As shown in Supplementary Table 5, our results have demonstrated that the TOF (at −0.2 V vs. RHE) of WS$_2$ MSLs is 0.739 s$^{-1}$, much larger than that of 1T'-WS$_2$ NSs (0.090 s$^{-1}$) and 2H-WS$_2$ NSs (0.078 s$^{-1}$), indicating the significantly enhanced intrinsic activity of WS$_2$ MSLs (see the Supplementary Note 2 for details on the calculation

of the TOF values). The excellent intrinsic activity of the WS$_2$ MSLs catalyst is likewise evidenced by its ECSA-normalized current density (Fig. 3d, Supplementary Fig. 25, Supplementary Table 3) and the comparison of mass activity with other WS$_2$-based electrocatalysts (Supplementary Table 6, Supplementary Note 3).

Excellent mass transfer performance has emerged as an essential factor to evaluate the property of high-efficiency electrocatalysts for HER. In the macro presentation, mass transfer is mainly the gas evolution and the contact between electrolytes and electrode surface, where it occurs at the solid–liquid–gas three-phase interface. Therefore, the wetting state of the electrode surface has become a significant factor to influence the whole mass transfer performance[55]. The contact angle (CA), as a parameter to measure the wettability at the intersection of gas, liquid, and solid, is one of the important criteria for evaluating wettability and even mass transfer performance[56]. Generally, solid surfaces with CAs < 90° are considered to be hydrophilicity, and those with CAs > 90° are hydrophobicity. Moreover, solid surfaces with CAs < 10° are considered to be superhydrophilicity[57–59] (see Supplementary Fig. 26 for detail). The CAs on the electrode surface of bare CFC and WS$_2$ MSLs-CFC are 127.1° and 9.1°, respectively, indicating the significant hydrophilicity of WS$_2$ MSLs (Fig. 4a), which benefits from the

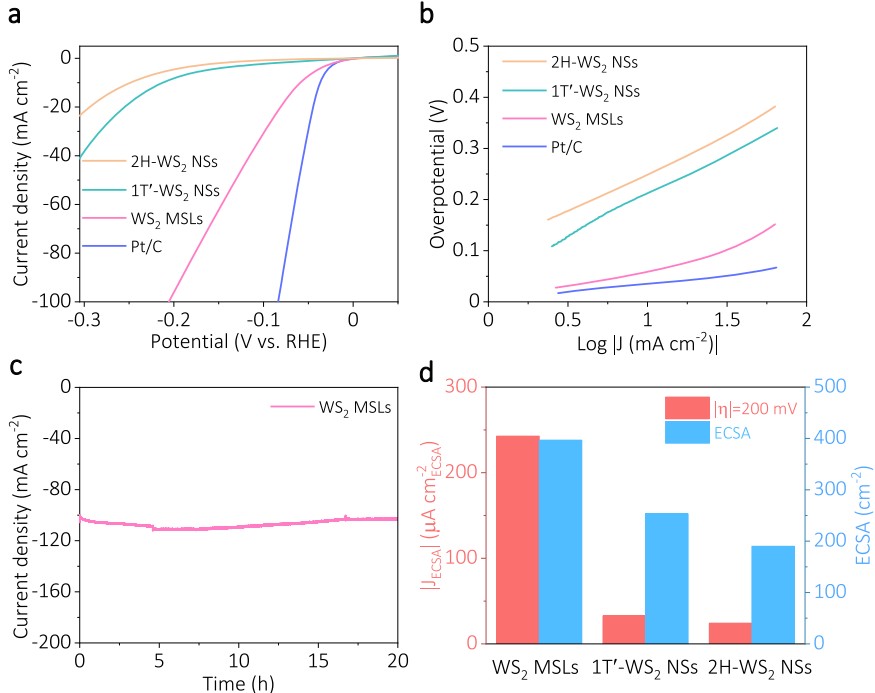

**Fig. 3 Electrocatalytic application of WS$_2$ MSLs in HER. a** Polarization curves of all catalysts with a scan rate of 10 mV s$^{-1}$ in Ar-bubbled 0.5 M H$_2$SO$_4$ (after iR correction, normalized by geometrical surface area, Geometric electrode area: 1 cm$^2$). **b** The corresponding Tafel curves for catalysts derived from (**a**). **c** Continuous HER recorded from synthesized WS$_2$ MSLs as working electrodes at a static potential of −0.2 V vs. RHE. **d** Comparison of the ECSA and $J_{ECSA}$ (at −0.2 V vs. RHE) of WS$_2$ MSLs, 1T'-WS$_2$ NSs, and 2H-WS$_2$ NSs.

unique micro–nanostructure and metallic phase of WS$_2$[60–62]. As shown in Supplementary Fig. 27, the hydrated cation preferentially adsorbs onto the 1T'-WS$_2$ surface, evidenced by more favored adsorption energy (−3.45 eV) as compared to slightly weaker adsorption energy of the 2H phase (−1.82 eV). In addition, the surface wettability of the electrode material under the electrolyte was investigated by measuring the CA of the hydrogen bubbles in the electrolyte (Fig. 4b), which further demonstrated the superhydrophilic and superaerophobic characteristics of the WS$_2$ MSLs.

Moreover, we investigate the visual behavior of as-generated gas bubbles releasing from the surface of WS$_2$ MSLs to demonstrate the morphological evolution of WS$_2$ nanoarray electrodes during the HER process. The circular three-phase contact line (TPCL) (white line) of as-formed bubbles on the electrode surface was easily cut into a discrete state by the intrinsically specific surface geometries with micro/nanoporous architecture (Fig. 4c). As expected, most of the hydrogen bubbles are smaller than 100 μm in diameter when they leave the surface of WS$_2$ MSLs as shown in Fig. 4d. Compared with the continuous TPCL on the ideal flat electrode surface (Supplementary Fig. 28 and Supplementary Movie 1), the cut three-phase contact line by nanoarray makes the big bubble split into small ones more naturally (Fig. 4c and Supplementary Movie 2), maintaining rapid and stable contact between the electrodes and electrolyte and deterring the formation of inactive sites (Fig. 4e).

The relatively smaller bubbles have much lower adhesion force with electrode surface and exhibit higher transportation velocity on cones[63]. As shown in Fig. 4f, the simplified stress analysis on a single bubble at the electrode surface indicated that the adhesion force ($F_a$) plays a pivotal part in gas bubble detachment. As expected, a small bubble adhesive force (10.4 ± 1.5 μN) was measured on the WS$_2$ MSLs–CFC surface underwater, accompanied with negligible shape change of the gas bubble (Fig. 4g and Supplementary Movie 3). The aerophilicity manifestation of bare

CFC in the adhesion measurements also further verified the superaerophobic property of the WS$_2$ MSLs (Supplementary Fig. 29 and Supplementary Movie 4). The obtained results definitely demonstrate that the superhydrophilic and superaerophobic characteristics of the unique micro/nano surface structure of WS$_2$ MSLs play vital roles in accelerating HER kinetics

**HER enhancement mechanism**. To reveal the origin of enhanced catalytic activity of WS$_2$ MSLs, the electronic properties of WS$_2$ MSLs were investigated by DFT in Supplementary Fig. 30. Obviously, the charge densities of WS$_2$ MSLs are bigger and clearer than that of normally stacked bilayer WS$_2$, indicating much stronger electron orbitals coupling in WS$_2$ MSLs[64]. Supplementary Fig. 31 shows various active sites of 1T@2H WS$_2$ nanobelts for catalytic HER. The optimized structural model of monolayer WS$_2$ plane consisting of 2H and 1T' phases (Supplementary Fig. 32a, b). shows that the WS$_2$ plane has undergone significant deformation with strain. The $\Delta G_H$ has been demonstrated to be a successful descriptor of the HER activity, where a value of $\Delta G_H$ closer to zero results in the higher activity. The $\Delta G_H$ was calculated for H adsorption on sites as marked in Supplementary Fig. 33. Obviously, the favorable sites for HER are distributed on the edges, polymorphs interface, and strained metallic phase surface. A linear relation between $\Delta G_H$ and p band center of S atom in Supplementary Fig. 32 indicated that the intrinsic activity of HER active sites was closely related to the p band center of S atom. The influence on $\Delta G_H$ of interesting MSLs was investigated by theoretical calculations. We performed DFT calculations on non-twisted bilayers WS$_2$ and twisted bilayers WS$_2$ with 14° to study the influence of $\Delta G_H$ induced by the twisted effect in Supplementary Figs. 34–38. Computational predictions for the MSLs effect on the HER activity indicated that the active sites of W-edge and S-edge of twisted bilayers WS$_2$ have much more appropriate $\Delta G_H$ compared with normal bilayers WS$_2$ in

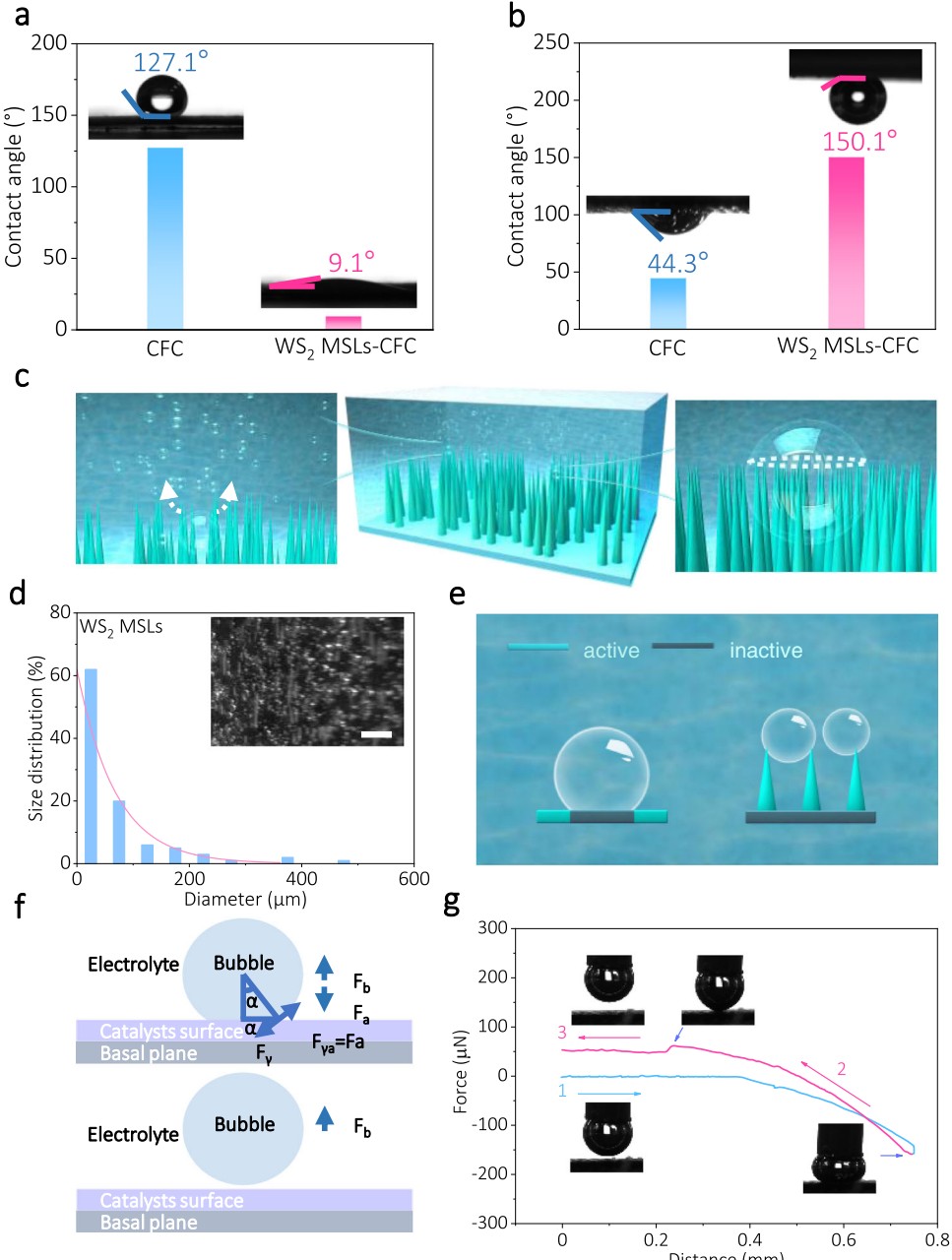

**Fig. 4 Mass transfer behavior research. a** Contact angles of an electrolyte droplet on the catalysts' surfaces. **b** Contact angles of a gas bubble on the catalyst surface under electrolyte. **c** Schematic illustration of how the conical nanoarrays surface morphology is affecting the bubble contacts and release. **d** Size distributions statistics of releasing bubbles on the surfaces of a WS$_2$ MSLs electrode, and (inert) digital photo is demonstrating the bubble releasing behaviors on the surface of WS$_2$ MSLs for HER. Scale bars, 1.0 mm. **e** Schematic illustration of bubble and catalysts contact. **f** Stress analysis of one single bubble on the surface of the catalyst. **g** Adhesive forces measurements of the gas bubbles on WS$_2$ MSLs–CFC surface.

Fig. 5. The atomic structure can fine-tune the electronic structure of the active sites by upshifting the d band center of W atoms and p band center of S atoms, which indicates the same trend of the enhanced hydrogen binding energy, thus promoting the HER performance (Supplementary Fig. 36c and Supplementary Fig. 38c). Accordingly, we evaluated the HER activity of 2H-WS$_2$ MSLs and 1T'-WS$_2$ MSLs using DFT by comparing $\Delta G_H$ for hydrogen adsorption at both basal planes (Supplementary Fig. 39). The $\Delta G_H$ close to zero at 1T'-WS$_2$ MSLs basal plane site (−0.24 eV) validates the HER activity of WS$_2$ MSLs relating to the base plane in this study. Therefore, the enhanced HER activity can be ascribed to the synergistic effect of phase and MSLs.

## Discussion

Through a combined theory-experiment approach, we identify and develop highly active WS$_2$ MSLs based on mechanical flexibility for the HER. We ascribe the total activity enhancement to a combination of electronic, geometric, superaerophobic, and superhydrophilic effects. The highly active under-coordinated sites at the edges, polymorphs interface, strained metallic phase surface of WS$_2$ MSLs are more active than those of non-twisted bilayers WS$_2$. $\Delta G_H$ is sensitive to the MSLs, which implies that engineering MSLs of WS$_2$ or other TMDs can be a route for their catalytic property engineering. This research extends twistronics and moiré fringe physics to HER catalysts and opens the

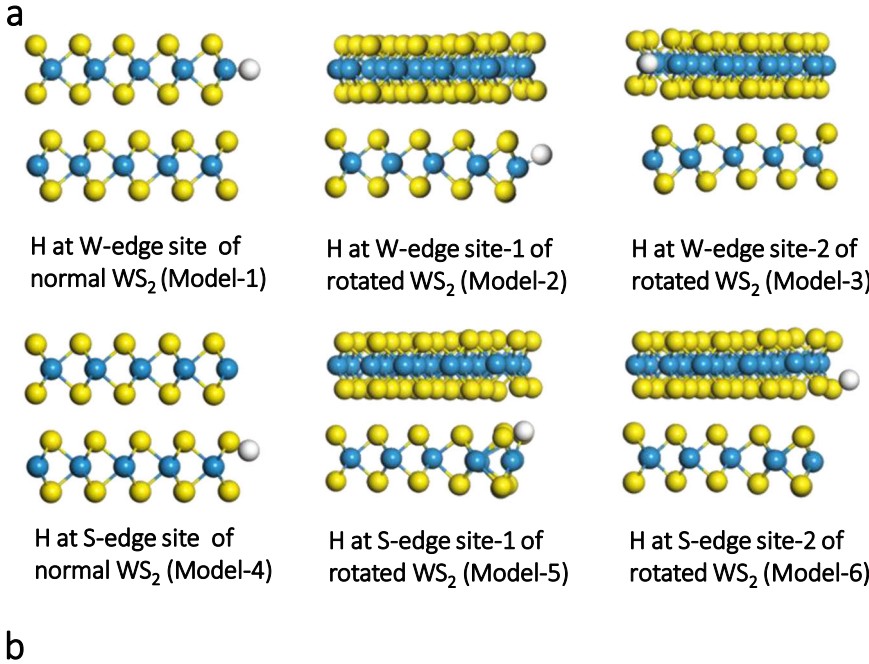

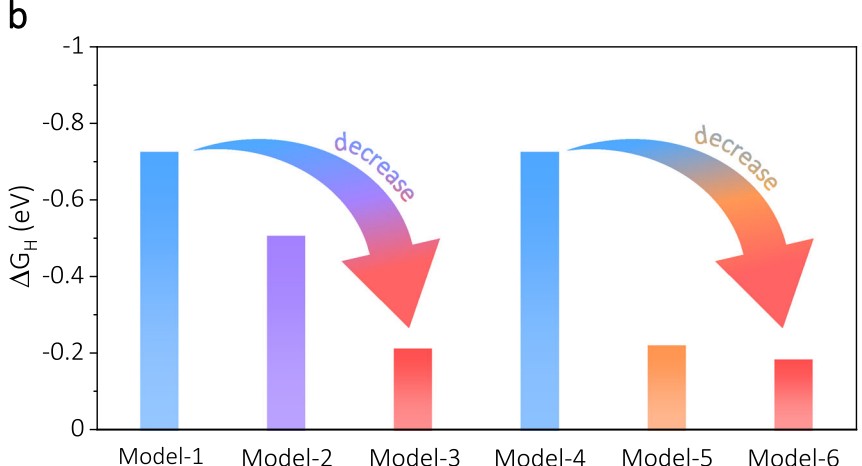

**Fig. 5 HER enhancement mechanism. a** Adsorption structures of H at the W-edge site of normal $WS_2$ (Model-1), W-edge site-1 of rotated $WS_2$ (Model-2), W-edge site-2 of rotated $WS_2$ (Model-3), S-edge site of normal $WS_2$ (Model-4), S-edge site-1 of rotated $WS_2$ (Model-5), and S-edge site-2 of rotated $WS_2$ (Model-6) (side view). Yellow, cyan, and white balls represent S, W, and adsorbed H atoms. **b** The changes (indicated by the arrows) of hydrogen adsorption free energy ($\Delta G_H$) values of various active sites corresponded to the different atomic structure models in (**a**).

possibility of designing the type of catalyst by topological physics engineering based on mechanical flexibility.

## Methods

### Materials synthesis

*Synthesis of $WS_2$ MSLs.* A facile one-step hydrothermal method was used to fabricate $WS_2$ MSLs. For the synthesis, 0.5 mmol $(NH_4)_{10}W_{12}O_{41}\cdot xH_2O$ and 30 mmol $CH_4N_2S$ were dispersed in 35 mL distilled water (60 °C) by sonication for 1 h. The hydrothermal reaction was carried out in a 45 mL Teflon-lined stainless-steel autoclave. The above mixture solution was transferred into the autoclave and maintained at 300 °C for 100 h. After cooled to room temperature gradually, the as-obtained product was centrifuged and dried in a vacuum at 60 °C.

*Synthesis of metallic 1T' phase dominated $WS_2$ nanosheets, referred to as 1T'-$WS_2$ NSs.* Typically, 0.2 mmol $(NH_4)_{10}W_{12}O_{41}\cdot xH_2O$ and 2.4 mmol thioureas were added to a 100 mL three-neck flask containing 40 mmol oleylamine (OM, 70%, Sigma-Aldrich) at room temperature. Then, vacuum the system for 5 min and inject of $N_2$ for 5 min. This process is cycled three times and carried out at 120 °C. The mixture solution was rapidly heated to 280 °C and vigorously stirred for 90 min under an $N_2$ atmosphere. After cooled to room temperature gradually, the black products were filtered and washed 5 times with cyclohexane and ethanol. The as-obtained 1T'-$WS_2$ nanosheets were centrifuged and dried in a vacuum at 60 °C.

*Synthesis of 2H phase $WS_2$ nanosheets referred to as 2H-$WS_2$ NSs.* For comparison, the 2H-$WS_2$ sample was prepared by heating the 1T'-$WS_2$ sample at 300 °C for 2 h in a vacuum.

*Material characterizations.* TEM images were measured by transmission electron microscope (Hitachi HT7700). High-resolution TEM images were operated at an acceleration voltage of 200 kV (FEI Talos F200X). SEM images were measured using an FE-SEM (S-4800). XRD patterns were recorded with an X-ray diffractometer (Bruker AXS D8 Advance A25), using Cu Kα radiation ($\lambda = 1.5406$ Å) over the range of $2\theta = 5.0$–$80.0°$. Data were collected using $2\theta$ scan step of 0.02° at a rate of 2° min$^{-1}$. XPS measurements were conducted using an X-ray photoelectron spectrometer (KRATOS Axis Supra). A 200 W Mg X-ray excitation was used. All the samples were analyzed with reference to adventitious carbon 1 s peak. Raman spectra were recorded by a confocal Raman microscope (inVia, Renishaw, England) equipped with a 532 nm He–Ne laser as an excitation source.

*Electrochemical measurements.* All the electrochemical experiments were carried out using a conventional three-electrode system on an Electrochemical Workstation (CS310, Wuhan Kesite Instrument Co., Ltd.). All electrochemical performance tests of samples were carried out on CFC (Phychemi (HK) Company Limited-W0S1010). A typical three-electrode configuration was used to investigate all samples' HER performance with an Ag/AgCl electrode and a graphite rod as the reference and counter electrodes, respectively. All the electrochemical measurements were conducted in Ar-bubbled 0.5 M $H_2SO_4$ electrolyte at room temperature. All potentials were referenced to the reversible hydrogen electrode (RHE). Before the

electrochemical test, the fresh as-prepared 1T'-WS$_2$ NSs product and 2H-WS$_2$ NSs were added into a 100 mL Erlenmeyer flask containing 3 mL thioglycolic acid and 50 mL ethanol, and vigorously stirred for 12 h under N$_2$ atmosphere to partially removing the surfactant molecules. After that, the acid-treated 1T'-WS$_2$ NSs were separated from the solution by centrifugation (8500 rpm, 10 min), washed twice with ethanol. The catalyst dispersion was prepared by mixing 5.0 mg of catalyst in an aqueous solution containing 20 μL of Nafion (5 wt%), 800 μL D.I. water, and 200 μL absolute ethanol, and the ultrasound-treated time was 45 min. Then, the sample dispersion (40 μL) was dropped onto the cleaned CFC surface (1 cm$^2$) (equivalent to 0.196 mg cm$^{-2}$) and dried overnight naturally. The linear sweep voltammetry was tested at the potential of −0.35 to 0.10 V vs. RHE with a scan rate of 10 mV s$^{-1}$. The sweep rate of 10 mV s$^{-1}$ we used is slow enough to build a steady-state electrode and thus the resulting polarization curve is reasonable to be used for kinetic analysis[65]. The overpotential ($\eta$) plotted as a function of log current (log $J$) to obtain a Tafel plot for evaluating the HER kinetics of the electrocatalyst. The Tafel slope ($b$) can be obtained by the calculation of the Tafel equation ($\eta = b \log (J) + a$). By extrapolating the linear region back to zero overpotential, the exchange current density ($j_0$) can be obtained from the Tafel plots in Fig. 3b. CV and EIS were performed to evaluate the electrochemical double-layer capacitance ($C_{dl}$) of the materials at non-Faradaic processes as the means of estimating the corresponding electrochemical active surface areas (ECSA). For CV measurements, a series of CV curves were performed at various scan rates (10–140 mV s$^{-1}$) in the 0.25–0.30 V vs. RHE region. The cathodic (○) and anodic (□) charging currents tested at 0.275 V (vs. RHE) plotted as a function of scan rate. The $C_{dl}$ value of the system is obtained by calculating the average of the absolute value of the fitted line slope. The $C_{dl}$ values were used to estimate the ECSA of catalysts. (Calculation details are provided in Supplementary Note 2). The chronoamperometry test was performed to measure the electrocatalyst's stability during catalysis. WS$_2$ MSLs, 1T'-WS$_2$, and 2H-WS$_2$ NSs-coated CFC (1 cm$^2$, catalyst loading 280 μg) were used as working electrodes to collect chronoamperometry data at a static overpotential of 0.2 V. The uncompensated resistance was measured by EIS. The EIS measurements were performed in the same configuration at 250 and −50 mV (vs. RHE) from 100 kHz to 0.1 Hz. The electrolyte resistance ($R_s$) was obtained by the fitted Nyquist plots and used for iR compensation by the equation of $E_{iR-corrected} = E_{original} - I \times R_s$[66,67]."

**Bubble adhesion force test.** The electrolyte used in all tests is Ar-bubbled 0.5 M H$_2$SO$_4$. The CAs between the electrolyte and electrode surface was tested by a KRUSS (DSA20) system in ambient air. The Dataphysics DCAT25 system was used to measure the CAs between the gas bubbles and electrode surface under the electrolyte. The adhesion and desorption process of bubbles through the system comes with high-speed camera capture.

## Data availability

The data that support the findings of this study are available from the corresponding authors upon reasonable request. All source data underlying Figs. 3a–d, 4a, b, d, g, and 5b are provided as a Source Data file.

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

## Acknowledgements

This work was financially supported by the National Funds for Distinguished Young Scientists (61825503), the Natural Science Foundation of China (51902101, 61775101, and 61804082), the Youth Natural Science Foundation of Hunan Province (2019JJ50044), Natural Science Foundation of Jiangsu Province (BK20201381), and Science Foundation of Nanjing University of Posts and Telecommunications (NY219144).

## Author contributions

Q.Z. and W.H. conceived the project. L.L.W. designed, supervised, and analyzed the whole project. L.B.X. carried out the materials synthesis and electrochemical test. L.L.W. and L.B.X. wrote the paper together. S.J.L. and W.W.Z. analyzed the data. All the authors contributed to the discussion during the whole project.

## Competing interests

The authors declare no competing interests.
