## [Peer Review File · Nature Communications]

REVIEWER COMMENTS

Reviewer #1 (Remarks to the Author):

In this manuscript, the author reported a MSLs material WS₂ by a facile one-step hydrothermal method. They also found this material has high HER activity. The study on the HER of WS₂ are not novel, but the MSLs material for HER is interesting. Therefore, I would recommend publication after major revisions. My specific comments are as follows.

1, The author called the WS₂ they synthesized as moire superlattices (MSL) material, but they didn't provide sufficient evidence to support it is the MSL material. The superlattice structure is not very obvious. Further experimental results are supposed to give.

2, They used the whole Figure 4 to explain the good hydrophilic and aerophobic characteristics from the morphology. However, the author needs to provide some intrinsic explanation for this specifically twistrionics WS₂ HER properties, not simply attribute to the morphology of material.

3, In the DFT calculation section, what is the basis of the author modeling the initial periodic superlattice periodic structure MSLs WS₂? How to prove the modeling geometry are consist with the experimental structure? Did the author fully relax the initial constructed structure to provide the optimization ? Did the author give any strain or else vectors to keep the structure not relax to the initial geometry? How does the author know the activity only occurs on the edge not the basic plane ? Anyway, the author need to give more intrinsic explanation from the electronic structure, not only calculated the adsorption energy of H⁺.

4, The author stressed the superhydrophilic of MSL WS₂ in Figure 4 contribute to the high HER. The proton lives in the hydrated states as the hydronium ion H₃O⁺ in aqueous electrolyte. That means the H₃O⁺ has super strong interaction with the material surface, but the high HER activity corresponds with the moderate adsorption energy of proton in Figure 5. This seems to be contradictory.

5, More very related references are supposed to be added, such as Nature, 579, 353-358, 2020 and Nature, 579, 339, 2020, etc.

6, In Figure 5 caption, it is supposed to be W, not Mo.

Reviewer #2 (Remarks to the Author):

In the work entitled "WS₂ moiré superlattices derived from mechanical flexibility for hydrogen evolution reaction", WS₂ moiré superlattices (MSLs) are synthesised, characterised and tested as hydrogen evolution reaction electrocatalysts. After carefully considering the manuscript, I have identified many areas where the data analysis is performed erroneously leading to potentially misleading conclusions. Given the high profile of Nature Communications, I would recommend that major revisions are required before considering this work as suitable for publication. Here is a list of my concerns:

- In the abstract, the authors mention "electron-catalytic activity", is this supposed to be electro-catalytic activity?

- Also in the abstract, the authors mention "ascribed to much more appropriate ΔG_H of twisted bilayers WS₂ active sites". Without context, "more appropriate" is meaningless. Also, the typo "active sits" has been repeated throughout the manuscript (perhaps unsurprising, given that this sentence has been copy/pasted from later in the manuscript).

- In Figure 1c the authors mention finite element calculations, but as far as I can tell they have not provided any details on the nature of these calculations?

- In Figure 1d, it is not clear to me what is actually being represented. Is this a schematic? Or is it based on calculations? The figure caption (and associated in-text discussion) is not very helpful in this regard.
- In Figure 2a/b, the authors present evidence of the WS₂ MSLs in the form of STEM images. However, presumably this was taken at one point across a macroscopic sample (e.g., in the electrochemical tests, the electrode seems to be on the cm² scale). It would be useful if the authors comment on how representative this image is? Also, is the 13.82° twist angle observed for all WS₂ MSLs? Or does each nanobelt/nanocone have a unique twist angle?
- In Figure 3a: What is the scan rate? Electrolyte? Geometric electrode area? Also, have these been iR corrected? None of this is mentioned in the figure caption.
- In Figure 3a (and SI), potential is spelled wrong on the x-axis.
- In Figure 3b, it appears as if the Tafel slope for the green and yellow traces has been estimated over a very narrow range of current densities (not even 1 order of magnitude). Why is this the case when the author have clearly measured the currents over a larger range of current densities (i.e., in Figure 3a)? Also, how can the authors be confident of their assignment of Tafel slopes over such a narrow range?
- My major concern with this work is how the data are normalised. Currents appear to be normalised to area (i.e., current density), but it is not clear to me what the area corresponds to or how it was measured. Is this the actual area of the active material? If yes, how was this estimated? Alternatively, if this is geometric area, then the statement "MSLs show superior catalytic activity for HER" is very misleading. Electrochemical activity can be benchmarked by measuring the exchange current density (j_0 , with units of mA cm⁻²) or the heterogeneous electron-transfer rate constant (k_0 , with units of cm s⁻¹). Due to the complexity of multi-electron processes, j_0 is almost always adopted for complex catalytic reactions such as the HER. In principle, j_0 can be estimated from the Tafel plots in Figure 3b (by extrapolating the linear region back to zero overpotential), with the caveat that the active electrode area must be known. If this is not known or it cannot be measured, then the difference in the curves shown in Figure 3a/b could EITHER be due to changes in activity (i.e., a legitimate difference in j_0) or could simply be due to differences in the roughness/porosity of the electrode. In principle, even a very 'inactive' catalyst could present a lower overpotential (hence higher apparent "activity") at a given geometric current density if the exposed surface area is high enough. Misleading statements about the "activity" of rough/porous electrodes is a major problem in the electrocatalysis literature, as it makes it very difficult to compare results from different labs. There have been many reviews that discuss this fact (e.g., 10.1021/cs500923c).
- In Figure 3a/b, Pt/C is presented but never mentioned/discussed in the main text.
- The form of analysis performed in Figure 3d assumes that the electrodes behave as "ideal" electrochemical capacitors (i.e., current scales linearly with scan rate). However, all three plots do not pass through the origin, (0,0), and some even show clear deviations from linearity (i.e., see the 1T'-WS₂ NSs). Consulting the source data (Supplementary Figure 14), all three electrodes exhibit asymmetric I-E plots (i.e., the reduction current is larger in magnitude than the oxidation current) that are somewhat sloped. This may suggest that a charge-transfer reaction (i.e., Faradaic process) may also be occurring in this potential range (i.e., at $E < 0.20$ V, where the baseline is clearly sloped). Is it appropriate to perform such a simple form analysis with such a complex electrode architecture? Perhaps the authors could investigate how closely the electrodes mimic an "ideal capacitor" with electrochemical impedance spectroscopy? In addition, In Figure 3d, there is no explanation as to what the units on the y-axis (DJ0.2V) actually means.
- The authors state: "The much smaller Tafel slopes of WS₂ MSLs (40 mV decade⁻¹) indicated that the kinetics of the electrochemical hydrogen evolution on WS₂ MSLs was much faster

than those of the 2H-WS2 NSs and 1T'-WS2 NSs (Fig. 3b).” Tafel slopes DO NOT indicate on kinetics, rather they can indicate on the reaction mechanism under some circumstances (i.e., for a well-defined, dimensionally stable electrode). As noted above, j_0 would be an indicator of kinetics, but this has not been calculated.

- The authors state: “Additionally, compared to 2H-WS2 and 1T'-WS2 NSs, the lower charge transfer resistance and rapidly electron transportation capability of the WS2 MSLs are confirmed by the electrochemical impedance spectroscopy (EIS) measurements (Supplementary Fig. 11).” However, they do not offer any justification/discussion as to why the EIS measurements indicate this. They do not even fit the spectra or include an equivalent circuit. Given the high porosity of the electrodes in question, interpreting the EIS spectra is not straightforward.

- The authors state: “The electrochemical double-layer capacitances (Cdl) were calculated to contrast the electrochemical surface area (ECSA) of 2H-WS2 NSs, 1T'-WS2 NSs and WS2 MSLs (Fig. 3d). The Cdl of WS2 MSLs (33.7 mF cm^{-2}) was much higher than that of 1T'-WS2 NSs (21.2 mF cm^{-2}) and 2H-WS2 NSs (7.2 mF cm^{-2}), indicating that the WS2 MSLs possessed more fully exposed active sites for electrochemical hydrogen evolution.” How were these values normalised? Geometric area? In principle, if the specific capacitance (C_{specific} in F cm^{-2}) of these materials was known, then the exposed surface area (or ECSA) could be estimated from these values (i.e., $\text{AECSA} = C_{\text{dl}} / C_{\text{specific}}$). The ECSA however does not necessarily indicate on the number of exposed “active sites”, as all sites, regardless of ‘activity’ (e.g., the basal and edge planes of 2H-WS2 plus the underlying carbon support) contributes to the non-faradaic current, whereas only certain sites (e.g., the edge plane of 2H-WS2) may dominate the HER catalysis. How was the carbon support corrected for when calculating the ECSA?

- The authors use the terms “superhydrophilic” and “superaerophobic” throughout. It would be useful if they provide a definition of these terms for the reader (i.e., what distinguishes hydrophilic from superhydrophilic?).

- In Figure 5, it is not clear to me what the numbers “1,2,3,4,5,6” refer to or what the arrows are indicating. It is also not clear why a 3D plot is necessary here? The pictures of the various active sites are also too small to see clearly, so overall it is very difficult to determine anything meaningful from this figure.

- The authors state “We ascribe the activity enhancement to a combination of electronic, geometric, superaerophobic and superhydrophilic effects.” Again, as stated above, in

electrochemistry “activity” refers to electron-transfer kinetics. This statement is misleading, as the aforementioned “superaerophobic and superhydrophilic effects” influence the mass transfer of bubbles, rather than enhancing electron-transfer kinetics

- The authors state “ ΔG_{H} is insensitive to the MSLs” which is counter to the argument that they presented in Figure 5 (e.g., “Computational predictions for the MSLs effect on the HER activity indicated that the active sites of W-edge and S-edge of twisted bilayers WS2 have much more appropriate ΔG_{H} compared with normal bilayers WS2 in Fig. 5.”)

- The “Electrochemical Measurements.” Section of the SI is not sufficiently detailed to enable the reader to repeat the measurements. For example, in the “supplementary methods section” the authors state “The full description of linear sweep voltammetry (LSV) and cyclic voltammetry (CV) tests have been shown in supplementary methods.” Also “The value of electrochemical double layer capacitance (Cdl) Electrochemical active surface areas (ECSA) was calculated by measuring CV curves of samples.” is very ambiguous. Further “Nyquist plots of three samples were measured in the frequency range from 100 Hz to 0.1 kHz at an open circuit potential of -350 mV.” According to Figure 3a of the main text, -350 mV is well beyond the onset potential of the HER on all considered electrodes. Therefore it is simply impossible that -350 mV could correspond to the “open circuit potential” of the catalysts.

Reviewer #3 (Remarks to the Author):

In this manuscript, the authors reported the synthesis of large-scale WS₂ Moiré superlattices (MSLs) through a one-pot hydrothermal approach, and demonstrated their catalytic hydrogen production performance. However, the evidence of Moiré superlattices is not sufficient and doesn't support the main theme of this article. Thus, the main claim in this work is weak. Upon a careful examination, I cannot recommend its publication in Nature Communications.

Some detailed comments are listed as follows:

- 1) HRTEM images in Figure 2a and 2b seems have been applied too much filters during the data recording and processing, and the images didn't show a clear Moiré period even in a small region.
- 2) From the XRD results in Fig. S5, we can see the 1T'-WS₂ NSs sample has a low crystallinity, to make a comparison, the standard PDF card should be presented in the same image.
- 3) The EIS measurements carried out at a large overpotential (-350 mV), which is not appropriate. The EIS measurements are better carry out at a small catalytic current region.
- 4) In line2, page 7. The "S-Mo-S" should be S-W-S.

Point-by-point response to the referees' comments

We sincerely thank the referees for carefully reviewing our manuscript and their valuable comments, which certainly help us improve our manuscript. We also thank the editor for giving us such an opportunity to address the comments and revise the manuscript. The changes in the revised manuscript have been highlighted in red for your review. The point-by-point responses are presented below.

Finally, we sincerely express our gratitude to the referee for all the constructive comments and suggestions, which really help us to improve our understanding of the relation between structures and properties of WS₂ MSLs. Following the referee's suggestions, we have conducted more experimental and theoretical analysis to provide more convincing evidences of WS₂ MSLs and correspondingly modify the theoretical models to reflect the most possible sample condition. We believe that the quality of the revised manuscript has reached the requirements of Nature Communications.

Reply to Referee 1 and revisions made accordingly:

In this manuscript, the author reported a MSLs material WS₂ by a facile one-step hydrothermal method. They also found this material has high HER activity. The study on the HER of WS₂ are not novel, but the MSLs material for HER is interesting. Therefore, I would recommend publication after major revisions. My specific comments are as follows.

Response: Thanks for the reviewer's positive evaluation of this work and suggestion. We have made revisions according to each comment, as summarized below.

1. The author called the WS₂ they synthesized as moiré superlattices (MSL) material, but they didn't provide sufficient evidence to support it is the MSL material. The superlattice structure is not very obvious. Further experimental results are supposed to

give.

Response: Thanks for the comments and suggestions from the reviewer. Moiré patterns can arise under two conditions, either when the two lattices have slightly different parameters or when identical lattices are twisted at an angle θ with respect to each other. HRTEM is widely used to characterize moiré patterns in 2D materials. In order to demonstrate that the synthesized WS₂ was moiré superlattices (MSL) material, we provide HRTEM images and the simulated HRTEM image of WS₂ as the sufficient evidence to support that it is the MSL material. We collected low-magnified TEM images and estimated the moiré superlattices as shown in **Figure R1**. Low-magnified TEM image of WS₂ MSL in **Figure R1a** exhibits a well-arranged hexagonal lattice structure which is attributed to the twist of bilayer WS₂ with a twisted angle. As can be seen, moiré superlattices are found throughout the measured region. The corresponding FFT patterns contain double sets of 6-fold symmetry diffraction spots. According to the measurement of the splitting spots in the FFT patterns, the misorientation angle of $\sim 13.8^\circ$ could be calculated from the fast Fourier transformed (FFT) images as shown in **Figure R1b**. Herein, the twisted angle θ also could be obtained via the formula: $\theta = 2\arcsin a/\lambda$, where $a = 0.322$ nm is the lattice constant of WS₂ and $\lambda \approx 1.34$ nm is the moiré wavelength depicted in **Figure R1c** (*Nature* **2018**, 556, 80–84). The significant honeycomb-structured Moiré pattern in **Figure R1c** is consistent with the simulated HRTEM images of WS₂ MSL (**Figure R1d**).

Figure R1. (a) Low-magnified TEM image of WS₂ MSLs. (b) The corresponding FFT pattern of (a). (c) HRTEM image of WS₂ MSLs. (d) Simulated HRTEM image of WS₂ MSLs.

To address the reviewer's concern, we replaced some typical images with regular hexagonal MSLs domains in **Figure 2a, b** as sufficient evidence to support that it is the MSL material (page 7 in the revised manuscript). The relevant discussions have been added into the revised manuscript (Page 7, 2–4; page 8, line 5–11).

2. They used the whole Figure 4 to explain the good hydrophilic and aerophobic characteristics from the morphology. However, the author needs to provide some intrinsic explanation for this specifically twistrionics WS₂ HER properties, not simply attribute to the morphology of material.

Response: Thanks for the comments and suggestions from the reviewer. The two primary categories of activity measurements are “total electrode” activity (i.e., geometric electrode area-normalized measurements) and “intrinsic” activity (i.e., per-site turnover frequency, TOF). Total electrode activity measurements are useful for practical device performance comparisons, but they are not ideal for fundamental studies of novel catalyst materials because they do not reveal the physical or chemical origins of an electrode’s activity. Intrinsic activity measurements provide the activity of the catalyst on a per-site basis, and therefore contribute to the molecular-level structure-property-function relationships necessary to guide catalyst development (*ACS Catal.* **2014**, *4*, 3957–3971).

The prominent “total electrode” activity of WS₂ MSLs is due to the systematic optimization of electronic structure and geometric structure principally—that is, high “intrinsic” activity, abundant electrocatalytic active sites, excellent conductivity, and the hydrophilic and aerophobic characteristics of electrode surface for fast mass transfer. First, enhancing the intrinsic activity of catalytic sites in electrocatalysts requires an optimization of the Gibbs free energy of adsorption for reactants to speed the rate-determining step in the overall reactions (**Figure 5**). Second, the enhanced conductivity of electrode materials is significant to enable fast charge transfer, a requirement for which metallic 1T/1T' phases have demonstrated superior ability (**Supplementary Figure 22**). Third, the increased density of electrocatalytic active sites in electrocatalysts allows for the maximization of limited electrode surface area—porous nanostructures have a proven ability to facilitate increased surface area. **Figure 3d** shows HER activity normalized for the electrochemical active surface area. Finally, the optimization of mass transfer properties has a tremendous effect on the ultimate efficiency of electrocatalysts (**Figure 4**). Hydrophilic and aerophobic behavior would allow greater contact between electrode and electrolyte, and thus promote efficient mass transfer (*Nat. Commun.* **2018**, *9*, 2452).

Taking the reviewer’s suggestions seriously, the first-principles method was used to study the twist effect of bilayer WS₂. The superlattice with $\theta = 13.2^\circ$ was selected for

further calculations of the total potentials in moiré superlattices. As shown in **Figure R2**, obviously, compared with the referenced bilayer structure, the potential barrier between two adjacent layers in the moiré superlattice has much smaller height and width, implying the increased interlayer coupling between electron orbitals. Because of the reduced potential barrier and increased interlayer coupling in the WS₂ moiré superlattice, electrons can transfer easily from the conductive substrate to the active sites, thus leading to superior HER electrocatalytic performance.

Figure R2. Total potentials for the superlattice with $\theta = 0$ and 13.2° .

According to the reviewer's suggestion, we have included the above discussion into new version as intrinsic explanation for this specifically twistrionics WS₂ HER properties in our case. We added the **Figure R2** as a new Supplementary **Figure 23** in Supplementary Information. The relevant discussions have been added into the revised manuscript (Page 13, line 20–22; page 14, line 1–5).

3. In the DFT calculation section, what is the basis of the author modeling the initial periodic superlattice periodic structure MSLs WS₂? How to prove the modeling geometry are consist with the experimental structure? Did the author fully relax the initial constructed structure to provide the optimization? Did the author give any strain or else vectors to keep the structure not relax to the initial geometry? How does the author know the activity only occurs on the edge not the basic plane? Anyway, the

author needs to give more intrinsic explanation from the electronic structure, not only calculated the adsorption energy of H^+ .

Response: Thanks for the comments and suggestions from the reviewer.

(1) The twisted bilayer WS_2 MSLs can be formed when the two identical monolayers are stacked with a relative twist angle of θ . The lattice structure of the twisted bilayer can be commensurate, such that the twisted bilayer forms a crystallographic superlattice whose superlattice period is determined by θ (*ACS Nano* **2017**, *11*, 11714–11723). Additionally, moiré patterns of twisted bilayer can also introduce a periodic potential acting on the interlayer coupling between two monolayers of the twisted bilayer, which results in the formation of a moiré superlattice. The period of moiré superlattice can be equal to or smaller than that of the crystallographic superlattice in the twisted bilayer. For the rotating model, we fixed a W atom and rotated the whole layer atom by a certain angle. The optimized structures of bilayer 2H- WS_2 and 1T/1T'- WS_2 with the moiré unit cell were shown in **Figure R3**.

Figure R3. (a) The ball-and-stick schematics of rotating way for 2H- WS_2 . (b, c) Top

and side view of the optimized structures of bilayer 2H-WS₂ with an interlayer twist by a rotation angle 13.2°. (d) The ball-and-stick schematics of rotating way for 1T-WS₂. (e, f) Top and side view of the optimized structures of bilayer 1T-WS₂ with an interlayer twist by a rotation angle 13.2°. The dashed (red) parallelogram corresponds to the moiré unit cell.

(2) We build models with the rotational stacking faults of 13.2°, which are the calculable models closest to the rotational stacking faults of 13.8° obtained by the experiment. We also show excellent agreement between the experimental HRTEM image with the simulated HRTEM image, as shown in **Figure R4**. These results prove that the Moiré patterns observed in the experimental HRTEM images are indeed consistent with modeling geometry. We added the **Figure R4** as a new **Supplementary Figure 5** in Supplementary Information. The relevant discussions have been added into the revised manuscript (Page 9, line 4–5).

Figure R4. (a) HRTEM image of WS₂ MSLs. (b) Simulated HRTEM image of WS₂ MSLs.

Although it is still a great challenge for theoretical calculation to provide a complete description for the real and complicated experimental system, especially for the surface electrocatalysis, the theoretical analysis based on simplified model system still can help understand the surface reaction mechanism (*PNAS* **2011**, *108*, 937–943).

(3) We confirmed that the structures are indeed relaxed fully, i.e., both atomic positions and lattice constants are relaxed, to achieve a fully optimized stable structure. The structural optimization was performed until the total energy difference was less than 10^{-5} eV. In order to investigate the minimum energy path of S atoms in the structural phase transition of WS₂ stacking layer structures, the NEB method, as implemented in VASP, was used (*J. Am. Chem. Soc.* **2008**, *130*, 16739–16744; *Phys. Rev. B* **1996**, *54*, 11169; *Comput. Mater. Sci.* **1996**, *6*, 15). The generalized gradient approximation (GGA) with the Perdew, Burke and Ernzerhof (PBE) functional (*Phys. Rev. Lett.* **1996**, *77*, 3865) was used to treat the exchange-correlation interaction between electrons, and the electron-ion interactions were treated in projector augmented wave (PAW) formalism (*Phys. Rev. B* **1999**, *59*, 1758). A vacuum region larger than 15 Å and perpendicular to the WS₂ sheets (along the c axis) was applied to avoid the interaction between the sheets in neighboring cells caused by the periodic boundary condition. In our calculation, a kinetic-energy cutoff for plane-wave expansion was set to 500 eV. All atoms in each unit cell were fully relaxed until the force on each atom was less than 0.005 eV/Å. Electronic energy minimization was performed with a tolerance of 10^{-6} eV. The DFT-D3 approach (*J. Chem. Phys.* **2010**, *132*, 154104) was used in order to take into account the effect of the van der Waals interaction. In order to test the HER activity on the edge sites of WS₂ MSLs, we modeled a nanoribbon structure with a vacuum of 15 Å in the y-direction. Two kinds of edge structures are created by the above procedure, including the one that ends with the exposed W atoms and the other ending with S only. We have added the details of DFT calculations in the **Supplementary Note 4. Table R1** lists the binding energy per WS₂ unit and the interlayer distance for different system. Comparing to the AA stacking, the twisted bilayer is energetically more favorable, as evidenced by the reduction in binding energy and the decrease in interlayer distance.

Table R1. Interlayer misorientation (θ), interlayer distance (d_{eq}), and binding energy (BE) per WS₂ unit for twisted bilayer WS₂ structures from our calculations.

System	θ (deg)	BE (meV)	d_{eq} (Å)
--------	----------------	----------	--------------

2H – 2H	0	–66.24	3.71
2H – 2H	13.2°	–73.61	3.59
1T' – 1T'	13.2°	–78.47	3.25

(4) There was no strain given to keep the structure not relax to the initial geometry. The twisted bilayers were modeled using accidental angular commensurations. In a hexagonal lattice whose basis vector is \mathbf{a}_1 , \mathbf{a}_2 , a skewed supercell with basis vector $(n\mathbf{a}_1 + m\mathbf{a}_2)$ has a corresponding skewed angle. We have added these data as a new **Supplementary Figure 4**, and added the relevant discussions to the revised manuscript (Page 9, lines 1–3).

Figure R5. Schematic plot of the skewed supercell in a hexagonal lattice. The skewed angle θ_1 or θ_2 is defined as the angle between the basis vector $(n\mathbf{a}_1 + m\mathbf{a}_2)$ and the zigzag direction. A pair of such skewed supercell with either exact or close matching of their lateral periodicity can become commensurate by rotating one relative to the other. The formed twisted bilayer sheet has a rotation angle of $\theta = \theta_1 - \theta_2$.

(5) Computational predictions for the MSLs effect on the HER activity indicated that the active sites of W-edge and S-edge of twisted bilayers WS_2 have much more appropriate ΔG_{H} compared with normal bilayers WS_2 in **Figure 5**. Accordingly, we evaluated the HER activity of 1T'– WS_2 MSLs using DFT by comparing the Gibbs free energy (ΔG_{H}) for hydrogen adsorption at both basal plane (**Figure R6**). The ΔG_{H} close

to zero at 1T'-WS₂ MSLs basal plane site (−0.24 eV) validates the HER activity of WS₂ MSLs relating to the base plane in this study. We added the **Figure R6** as a new **Supplementary Figure 38** in Supplementary Information. The relevant discussions have been added into the revised manuscript (Page 20, line 14–18).

Figure R6. (a) Atomic models of hydrogen absorbed at the at the basal plane of 2H-WS₂ MSLs and 1T'-WS₂ MSLs, (b) Calculated Gibbs free energy (ΔG_H) diagram for hydrogen binding at the basal plane of 2H-WS₂ MSLs basal plane and 1T'-WS₂ MSLs.

(6) We have explored the electronic properties of twisted bilayers of WS₂/WS₂ within the framework of density functional theory. Our calculations show that the electronic structures of twisted bilayers of WS₂/WS₂ are different from those of the non-twisted ones. The charge density difference of twisted bilayer WS₂ is plotted in **Figure R7**, which is defined as $\rho_{\text{diff}} = \rho_{\text{BL}} - \rho_{\text{TOP}} - \rho_{\text{BOT}}$. For all the stacked bilayers there is a tiny but still clear charge accumulation in the middle plane between the two monolayers. The charge depletion occurs at the innermost two neighboring S planes. The spatial redistribution of charge density has a more delocalized character for 1T'-1T' MSLs. We added the **Figure R7** as a new **Supplementary Figure 29** in Supplementary Information. The relevant discussions have been added into the revised manuscript

(Page 19, line 8–16).

Figure R7. Spatial map of charge density difference for the twisted bilayer WS_2 (calculated as $\rho_{\text{diff}} = \rho_{\text{BL}} - \rho_{\text{TOP}} - \rho_{\text{BOT}}$). Regions of electron accumulation and depletion are denoted by red and blue lobes, respectively. Isosurface $\rho = 2 \times 10^{-4}$ e/bohr.

4. The author stressed the superhydrophilic of MSL WS_2 in Figure 4 contribute to the high HER. The proton lives in the hydrated states as the hydronium ion H_3O^+ in aqueous electrolyte. That means the H_3O^+ has super strong interaction with the material surface, but the high HER activity corresponds with the moderate adsorption energy of proton in Figure 5. This seems to be contradictory.

Response: Thanks for the comments from the reviewer. We appreciate the valuable comments and are pleased to clarify this issue. Two surface models through trigonal prismatic and octahedral coordination of W atoms were built to simulate 2H- WS_2 MSLs and 1T'- WS_2 MSLs, respectively. As shown in **Figure R5**, the hydrated cation preferentially adsorbs onto the surface of the 1T' phase, evidenced by a much lower adsorption energy (-3.45 eV) as compared to that of the 2H phase (-1.82 eV). Actually, the 1T' phase is formed due to the filling of electrons to the d orbitals of the W atom in the 2H phase. This is to say, as compared to 2H, the 1T' phase is an electron-rich phase

that exhibits a high affinity to the positively charged ions in the electrolyte solution and results in enhanced adsorption. The ΔG_H close to zero at 1T'-WS₂ MSLs basal plane site (−0.24 eV) validates the high HER activity of WS₂ MSLs in this study (**Figure R6**). Wettability is a function of the specific free energy for any given surface (i.e., higher surface free energy causes increased wettability) (*ACS Nano* **2016**, *10*, 9145–9155). It has also been reported that active edge sites with the moderate adsorption energy of proton could also act as hydrophilic species (*Nano Lett.* **2014**, *14*, 4314–432). So, it is reasonable that WS₂ MSLs have the moderate adsorption energy of proton while simultaneously providing superhydrophilic mediated through 1T' phase and active edges. We added the **Figure R8** as a new **Supplementary Figure 26** in Supplementary Information. The relevant discussions have been added into the revised manuscript (Page 16, line 19–22).

Figure R8. The adsorption of hydrated cation onto the surface of 2H-WS₂ MSLs and 1T'-WS₂ MSLs, respectively.

5, More very related references are supposed to be added, such as *Nature*, 579, 353–358, 2020 and *Nature*, 579, 339, 2020, etc.

Response: Thank the reviewer for providing us the literature, and it has been cited in the revised manuscript as Reference 19 and 20 (Page 24, line 8–11).

19. Regan EC, et al. Mott and generalized Wigner crystal states in WSe₂/WS₂ moiré

superlattices. *Nature* **579**, 359–363 (2020).

20. Tang Y, et al. Simulation of Hubbard model physics in WSe₂/WS₂ moiré superlattices. *Nature* **579**, 353–358 (2020).

6, In Figure 5 caption, it is supposed to be W, not Mo.

Response: Thanks for the reviewer's help. This mistake has been corrected in the revised manuscript (Page 19, line 4).

Reply to Referee 2 and revisions made accordingly:

In the work entitled “WS₂ moiré superlattices derived from mechanical flexibility for hydrogen evolution reaction”, WS₂ moiré superlattices (MSLs) are synthesised, characterised and tested as hydrogen evolution reaction electrocatalysts. After carefully considering the manuscript, I have identified many areas where the data analysis is performed erroneously leading to potentially misleading conclusions. Given the high profile of Nature Communications, I would recommend that major revisions are required before considering this work as suitable for publication. Here is a list of my concerns:

Response: We thank Reviewer 2 for carefully reviewing our manuscript and the raised valuable comments. The questions and suggestions raised by Reviewer 2 are extremely important and helpful for improving the quality of our work. As will be shown below, we have carried out a series of new measurements following Reviewer 2’s comments. According to lots of articles by *Thomas F. Jaramillo* et al, all the data were standardized and analyzed to ensure the accuracy of the conclusion in the revised manuscript. The main conclusion of our manuscript is further strengthened, and we believe the quality of the paper is significantly improved.

1. In the abstract, the authors mention “electron-catalytic activity”, is this supposed to be electro-catalytic activity?

Response: Thanks for the comments from the reviewer. According to the reviewer’s suggestion, after careful consideration, we believe that the “electrocatalytic activity” in our article is more accurate. We have changed “electron-catalytic activity” to “electrocatalytic activity” in the revised manuscript (Page 2, line 10).

2. Also in the abstract, the authors mention “ascribed to much more appropriate ΔG_H of twisted bilayers WS₂ active sits”. Without context, “more appropriate” is meaningless. Also, the typo “active sits” has been repeated throughout the manuscript (perhaps

unsurprising, given that this sentence has been copy/pasted from later in the manuscript).

Response: Thanks for the comments and suggestions from the reviewer. We have used “ascribed to appropriate ΔG_H of twisted bilayers WS_2 active sites” instead of “ascribed to much more appropriate ΔG_H of twisted bilayers WS_2 active sites” in the revised manuscript (Page 2, line 11–12). Furthermore, we have checked the manuscript thoroughly and make the corresponding revisions. The typo “active sits” have been corrected in the revised manuscript.

3. In Figure 1c the authors mention finite element calculations, but as far as I can tell they have not provided any details on the nature of these calculations?

Response: Thanks for the comments from the reviewer. We have added the details of finite element calculations in the Supplementary Information (**Supplementary Note 5**, page 64–65, marked by red). The supplementary section was as follows:

For a mechanical perspective, the formation of a WS_2 nanocone under unbalanced external forces could be deemed as the WS_2 nanobelt’s screwing process, which can be numerically modeled by using the typically finite element method (FEM). **Table R2(a)** shows the designed three-dimensional FEM model, wherein the WS_2 nanobelt is set as an elastic body meshed with 57045 eight-node hexahedral elements. The density, Young’s modulus, and Poisson’s ratio of the WS_2 nanobelt are specified as 7.51 g cm^{-3} , 272 GPa, and 0.22, respectively (*Nano Lett.* **2014**, *14*, 5097–5103). Two steps (screwing and partial release) are specified to approximate the WS_2 nanobelt’s screwing process, wherein the calculated results (such as stress and strain distributions) in Step 1 are taken as the pre-state for Step 2. **Table R2(b)** and (d) specify, respectively, the mechanical boundary conditions (BCs) in Steps 1 and 2, and **Table R2(c)** and (e) show the calculated results (normalized stress distributions) of the screwed WS_2 nanobelt, respectively.

Table R2. FEM-based modeling of the WS_2 nanobelt’s screwing process.

4. In Figure 1d, it is not clear to me what is actually being represented. Is this a schematic? Or is it based on calculations? The figure caption (and associated in-text discussion) is not very helpful in this regard.

Response: Thanks for the comments from the reviewer. **Figure 1d** was replaced to show the schematic diagram of moiré superlattices formed by S-W-S layer slipping. The corresponding associated in-text discussion was also changed (Page 6, line 19–20).

Figure R9. Morphological and structural characterizations. (a) FESEM image of the as-prepared WS₂ nanoarrays. Scale bar, 2 μ m. (b) STEM image of single screwed WS₂ nanobelt. Scale bar, 300 nm. (c) Finite-element calculations of strain in a nanocone. (d) Schematic diagram of moiré superlattices formed by S-W-S layer slipping.

5. In Figure 2a/b, the authors present evidence of the WS₂ MSLs in the form of STEM images. However, presumably this was taken at one point across a macroscopic sample (e.g., in the electrochemical tests, the electrode seems to be on the cm² scale). It would be useful if the authors comment on how representative this image is? Also, is the 13.82° twist angle observed for all WS₂ MSLs? Or does each nanobelt/nanocone have a unique twist angle?

Response: Thanks for the comments from the reviewer. We think that there are too many factors (temperature and pressure, etc) that can induce errors during the synthetic process, and it is almost impossible to ensure an identical twist angle of every nanobelts. However, the synthesized WS₂ nanocones in this work are very uniform. We collected

ten HRTEM images and their corresponding FFT images taken from different WS₂ nanocones to ensure the twist angle in **Figure R10**. All the value of the twist angles are in the range of 13° to 14°. So, all the nanocones have the similar twist angles. We added the **Figure R10** as a new **Supplementary Figure 3** in Supplementary Information. The relevant discussions have been added into the revised manuscript (Page 8, line 21–22; page 9, line 1).

Figure R10. Examples of Moiré pattern using HRTEM images randomly selected from different WS₂ nanocones and their corresponding FFT images.

6. In Figure 3a: What is the scan rate? Electrolyte? Geometric electrode area? Also, have these been iR corrected? None of this is mentioned in the figure caption.

Response: Thanks for the comments from the reviewer. Linear sweep voltammetry (LSV) was conducted at a scan rate of 10 mV s⁻¹ in Ar-bubbled 0.5 M H₂SO₄ electrolyte at room temperature (after iR correction). The sweep rate of 10 mV s⁻¹ we used is slow enough to build a steady state electrode and thus the resulting polarization curve is reasonably to be used for kinetic analysis. All electrochemical performance tests of samples were carried out on carbon fiber cloth (CFC) (Geometric electrode area: 1 cm²). We have added the test parameters and conditions in the part of **Figure 3** caption (page 11, line 3–4). For your convenience, the related corrections are as follows:

Figure R11. Electrocatalytic application of WS₂ MSLs in HER. (a) Polarization curves

of all catalysts with a scan rate of 10 mV s^{-1} in Ar-bubbled $0.5 \text{ M H}_2\text{SO}_4$ (after iR correction, normalized by geometrical surface area, Geometric electrode area: 1 cm^2). (b) The corresponding Tafel curves for catalysts derived from (a). (c) Continuous HER recorded from synthesized WS_2 MSLs as working electrodes at a static overpotential of 0.2 V vs. RHE . (d) Comparison of the ECSA and J_{ECSA} (at -0.2 V vs. RHE) of (I) WS_2 MSLs, (II) 1T'- WS_2 NSs, (III) 2H- WS_2 NSs.

Moreover, a more detailed description was presented in the electrochemical test section (Supplementary Information, page 4–6, marked by red). As shown in **Figure R12**, we also put all raw data without iR corrections and iR corrected data in the revised SI to give comprehensive information (**Supplementary Figure 13**).

Figure R12. Polarization curves of the different WS_2 -based catalysts. All the samples were tested in Ar-bubbled $0.5 \text{ M H}_2\text{SO}_4$ solutions. Curves both with and without iR corrections are provided for each sample.

7. In Figure 3a (and SI), potential is spelled wrong on the x-axis.

Response: Thanks for the comments from the reviewer. We have changed the “potencial” to “potential” in **Figure 3a** (and SI).

The above-mentioned **Figure 3a** and **Supplementary Figure 14–17** has been revised as follows:

Figure R13 (Corrected Figure 3a). Electrocatalytic application of WS₂ MSLs in HER. (a) Polarization curves of all catalysts with a scan rate of 10 mV s⁻¹ in Ar-bubbled 0.5 M H₂SO₄ (after iR correction, normalized by geometrical surface area, Geometric electrode area: 1 cm²).

Figure R14 (Supplementary Figure 19). C_{dl} measurements of electrocatalytic material: WS₂ MSLs sample.

Figure R15 (Supplementary Figure 20). C_{dl} measurements of electrocatalytic material: 1T'-WS₂ NSs sample.

Figure R16 (Supplementary Figure 21). C_{dl} measurements of electrocatalytic material: 2H-WS₂ NSs sample.

Figure R17 (Supplementary Figure 18). C_{dl} measurements of base material: bare CFC.

9. My major concern with this work is how the data are normalised. Currents appear to be normalised to area (i.e., current density), but it is not clear to me what the area corresponds to or how it was measured. Is this the actual area of the active material? If yes, how was this estimated? Alternatively, if this is geometric area, then the statement “MSLs show superior catalytic activity for HER” is very misleading. Electrochemical activity can be benchmarked by measuring the exchange current density (j_0 , with units of mA cm^{-2}) or the heterogeneous electron-transfer rate constant (k^0 , with units of cm s^{-1}). Due to the complexity of multi-electron processes, j_0 is almost always adopted for complex catalytic reactions such as the HER. In principle, j_0 can be estimated from the Tafel plots in Figure 3b (by extrapolating the linear region back to zero overpotential), with the caveat that the active electrode area must be known. If this is not known or it cannot be measured, then the difference in the curves shown in Figure 3a/b could EITHER be due to changes in activity (i.e., a legitimate difference in j_0) or could simply be due to differences in the roughness/porosity of the electrode. In principle, even a very ‘inactive’ catalyst could present a lower overpotential (hence higher apparent “activity”) at a given geometric current density if the exposed surface area is high enough. Misleading statements about the “activity” of rough/porous electrodes is a major problem in the electrocatalysis literature, as it makes it very difficult to compare results from different labs. There have been many reviews that discuss this fact (e.g.,

Response: Thanks for the comments and suggestions from the reviewer. Thank the reviewer for providing us the literature, we have benefited a lot from it. It has been cited in the revised manuscript as Reference 51 (Page 26, line 27–29).

In our electrochemical test, the loading amount of the WS₂ MSLs catalyts was 0.196 mg cm⁻². For comparison, different WS₂ samples (1T'-WS₂ NSs, 2H-WS₂ NSs) have kept the same loading amounts of 0.196 mg cm⁻² under the same test conditions. Furthermore, the current is normalized to the superficial geometric electrode area to compare the total geometric activity of tested materials (*ACS Energy Lett.* **2016**, *1*, 589–594). The electrochemical test results on different WS₂ samples demonstrated that WS₂ MSLs have higher total geometric activity. Provided the electrocatalysts exhibited the similar catalytic active sites and centers, the enhanced specific surface area (roughness/porosity) can lead to the better electrocatalytic performance (*ACS Catal.* **2014**, *4*, 3957–3971; *ACS Catal.* **2012**, *2*, 1916–1923). As the reviewer states, “This makes even a very ‘inactive’ catalyst could present a lower overpotential (hence higher apparent “activity”) at a given geometric current density if the exposed surface area is high enough.”

For HER, turnover frequency (TOF) and current density normalized by electrochemically active surface area (ECSA) can be employed to reflect the intrinsic activity of the active sites in catalyts (*Chem. Rev.* **1995**, *95*, 661–666; *Adv. Energy Mater.* **2016**, *6*, 1501758; *ACS Catal.* **2014**, *4*, 3957–3971; *ACS Catal.* **2012**, *2*, 1916–1923; *Energy Environ. Sci.* **2015**, *8*, 3022–3029). Given the difficulties of determining the nature and quantity of catalytically active sites on the surface that are needed to calculate a true TOF, particularly for such rough electrodes as in the case of this study, here we report the average turn-over frequency (TOF_{avg}) by normalizing to a per surface site basis for each catalyst, assuming all surface atoms (metal and S atoms) are active. The number of surface sites was inferred from the ECSA (see the **Supplementary Note 2** for details on the calculation of the TOF values). As shown in **Table R3**, our results have demonstrated that the TOF (at –0.2 V vs. RHE) of WS₂ MSLs is 0.739 s⁻¹, much

larger than that of 1T'-WS₂ NSs (0.090 s⁻¹) and 2H-WS₂ NSs (0.078 s⁻¹).

To further demonstrate the enhanced intrinsic activity of WS₂ MSLs, the polarization curves were normalized to electrochemically active surface area (ECSA), which was derived from the double-layer capacitance (C_{dl}, **Figure R14–17**). As shown in **Figure R18** and **Table R4**, WS₂ MSLs still reveals substantially higher HER current density than that of 1T'-WS₂ NSs and 2H-WS₂ NSs at the same overpotential under the same measurement conditions.

From both TOF and the ECSA-normalized current density, it can be concluded that the WS₂ MSLs possess significantly enhanced intrinsic activity. It is important to emphasize that any use of J_{ECSA} values calculated from ECSA should be used as an approximate guide for comparing specific activity and should not be interpreted as an absolute reflection of turnover frequency, especially when comparing high-surface area and/or nanoporous films (*J. Am. Chem. Soc.* **2015**, *137*, 4347–4357).

In addition, we also calculated the activities per mass of WS₂ MSLs, and compared it with the activities per mass of WS₂-based electrocatalysts in other literature (**Table R5**). Remarkably, the current density of WS₂ MSLs at -300 mV is about 168.7 mA cm⁻² with a mass activity of 843.5 A g⁻¹, indicating the intrinsically promoted activity (Detailed calculation method was shown in the **Supplementary Note 3**).

Table R3. The TOF values of different WS₂ catalysts.

Catalyst	Overpotential (mV)	TOF (s ⁻¹)	Active site density (cm ⁻²)
WS ₂ MSLs	100	0.231	12.16 × 10¹⁷
	150	0.477	
	200	0.739	
1T'-WS ₂ NSs	100	0.022	8.26 × 10 ¹⁷
	150	0.045	
	200	0.090	

	100	0.014	
2H-WS ₂ NSs	150	0.033	
	200	0.078	5.47×10^{17}

Figure R18. (a) HER activity normalized for the electrochemical active surface area (ECSA). (b) Comparison of the ECSA and J_{ECSA} (at -0.2 V vs. RHE) of (I) WS₂ MSLs, (II) 1T'-WS₂ NSs, (III) 2H-WS₂ NSs.

Table R4. Electrochemical values analysis of different studied WS₂ samples.

Samples	j_0 [mA cm ⁻²]	Double-layer capacitance (C_{dl}) [mF cm ⁻²]	Electrochemical active surface area (ECSA) relative to WS ₂ MSLs	Current density normalized to ECSA (J_{ECSA}) ($\eta = -0.2$ V) [μ A cm ⁻²]
WS ₂ MSLs	0.848	23.8	1	242.68
1T'-WS ₂ NSs	0.394	15.2	0.64	32.75
2H-WS ₂ NSs	0.208	11.4	0.48	24.14

Table R5. Comparison of mass activity of various WS₂-based electrocatalysts.

Catalysts	Mass activity (A g ⁻¹)	Reference
-----------	------------------------------------	-----------

WS₂ MSLs	843.5 A g⁻¹ at -300 mV	
1T'-WS ₂ NSs	192.3 A g ⁻¹ at -300 mV	This work
2H-WS ₂ NSs	109.6 A g ⁻¹ at -300 mV	
As-exfoliated WS ₂ nanosheets	~92.8 A g ⁻¹ at -300 mV	Nat. Mater. 2013, 12, 850–855.
1T'-D WS ₂	~135.7 A g ⁻¹ at -300 mV	Nano Energy , 2018,
2H-WS ₂	~44.46 A g ⁻¹ at -300 mV	50, 176–181.
Compared WS ₂	~387.3 at -250 mV	Adv. Mater. 2018, 30, 1705509
WS ₂ composite (DMF)	~141.6 A g ⁻¹ at -300 mV	Adv. Funct. Mater. 2015, 25, 1127–1136

As shown in **Figure R19**, the exchange current densities (j_0) of the different samples were determined from the Tafel plots. The j_0 of 0.848 mA cm⁻² for WS₂ MSLs sample surpasses the values of 0.394 mA cm⁻² for 1T'-WS₂ NSs sample and 0.208 mA cm⁻² for 2H-WS₂ NSs sample. These results highlight the electrochemical activity of this WS₂ MSLs.

Figure R19. Exchange current density calculation. Calculated exchange current for different studied WS₂ samples by using extrapolation methods.

We have included these results in the Supplementary Information and the revised manuscript, **Figure 3d, Supplementary Table 3, 5, 6, Supplementary Figure 16, 24**, together with discussions. We have included the details about the calculation of the TOF values and ECSA values in the **Supplementary Note 1** and **Supplementary Note 2**, respectively. The relevant discussions have been added into the revised manuscript (Page 12, line 13–18; page 14, line 6–14).

51. Benck JD, Hellstern TR, Kibsgaard J, Chakthranont P, Jaramillo TF. Catalyzing the hydrogen evolution reaction (HER) with molybdenum sulfide nanomaterials. *ACS Catal.* **4**, 3957–3971 (2014).

10. In Figure 3a/b, Pt/C is presented but never mentioned/discussed in the main text.

Response: Thanks for the comments and suggestions from the reviewer. The relevant discussions have been added into the revised manuscript (Page 12, line 3–10).

11. The form of analysis performed in Figure 3d assumes that the electrodes behave as “ideal” electrochemical capacitors (i.e., current scales linearly with scan rate). However, all three plots do not pass through the origin, (0,0), and some even show clear deviations

from linearity (i.e., see the 1T'-WS₂ NSs). Consulting the source data (Supplementary Figure 14), all three electrodes exhibit asymmetric I-E plots (i.e., the reduction current is larger in magnitude than the oxidation current) that are somewhat sloped. This may suggest that a charge-transfer reaction (i.e., Faradaic process) may also be occurring in this potential range (i.e., at $E < 0.20$ V, where the baseline is clearly sloped). Is it appropriate to perform such a simple form analysis with such a complex electrode architecture? Perhaps the authors could investigate how closely the electrodes mimic an “ideal capacitor” with electrochemical impedance spectroscopy? In addition, In Figure 3d, there is no explanation as to what the units on the y-axis ($\Delta J_{0.2V}$) actually means.

Response: Thanks for the comments and suggestions from the reviewer. We measured the capacitive currents in a potential range where no faradic processes are observed, i.e. at 0.25–0.30 V vs. RHE. The cathodic and anodic charging currents measured at 0.275 V vs RHE plotted as a function of scan rate. The determined double-layer capacitance of the system is taken as the average of the absolute value of the slope of the linear fits to the data (Figure R20–23). The background capacitance of the bare CFC electrode was subtracted from the obtained double layer capacitance of all samples to compensate for the low substrate coverage (*Nat. Commun.* **2019**, *10*, 2650) (Figure R23).

Figure R20. Double-layer capacitance measurements for determining electrochemically-active surface area for WS₂ MSLs sample from voltammetry in Ar-

bubbled 0.5 M H₂SO₄. (a) Cyclic voltammograms were measured in a non-Faradaic region of the voltammogram at the following scan rate: 10, 20, 40, 60, 80, 100, 120, and 140 mV s⁻¹. The working electrode was held at each potential vertex for 10 s before the beginning the next sweep. All current is assumed to be due to capacitive charging. (b) The cathodic (○) and anodic (□) charging currents measured at 0.275 V vs. RHE plotted as a function of scan rate.

Figure R21. Double-layer capacitance measurements for determining electrochemically-active surface area for 1T'-WS₂ NSs sample from voltammetry in Ar-bubbled 0.5 M H₂SO₄. (a) Cyclic voltammograms were measured in a non-Faradaic region of the voltammogram at the following scan rate: 10, 20, 40, 60, 80, 100, 120, and 140 mV s⁻¹. The working electrode was held at each potential vertex for 10 s before the beginning the next sweep. All current is assumed to be due to capacitive charging. (b) The cathodic (○) and anodic (□) charging currents measured at 0.275 V vs. RHE plotted as a function of scan rate.

Figure R22. Double-layer capacitance measurements for determining electrochemically-active surface area for 2H-WS₂ NSs sample from voltammetry in Ar-bubbled 0.5 M H₂SO₄. (a) Cyclic voltammograms were measured in a non-Faradaic region of the voltammogram at the following scan rate: 10, 20, 40, 60, 80, 100, 120, and 140 mV s⁻¹. The working electrode was held at each potential vertex for 10 s before the beginning the next sweep. All current is assumed to be due to capacitive charging. (b) The cathodic (○) and anodic (□) charging currents measured at 0.275 V vs. RHE plotted as a function of scan rate.

Figure R23. Double-layer capacitance measurements for determining electrochemically-active surface area for the substrate material (bare CFC) from voltammetry in Ar-bubbled 0.5 M H₂SO₄. (a) Cyclic voltammograms were measured in a non-Faradaic region of the voltammogram at the following scan rate: 10, 20, 40, 60, 80, 100, 120, and 140 mV s⁻¹. The working electrode was held at each potential

vertex for 10 s before the beginning the next sweep. All current is assumed to be due to capacitive charging. (b) The cathodic (\circ) and anodic (\square) charging currents measured at 0.275 V vs. RHE plotted as a function of scan rate.

According to the reviewer's suggestion, the double-layer capacitance of the catalyst can also be determined by electrochemical impedance spectroscopy (EIS) in the non-Faraday region. EIS can be used to obtain the double layer capacitance by fitting the impedance response of the system at different frequencies to the Randles circuit. The solid lines are the fits to the data using the simplified Randles circuit shown in the inset of **Figure R24**. The determined double-layer capacitance of the WS₂ MSLs, 1T'-WS₂ NSs, and 2H-WS₂ NSs from the fitted data is 27.3 mF cm⁻², 15.1 mF cm⁻², 10.8 mF cm⁻², respectively (All fitting parameters are listed in **Table R6**). The double-layer capacitance we measured by EIS is within 15% of that measured from the scan rate-dependent CVs (**Table R7**) (*J. Am. Chem. Soc.* **2013**, *135*, 16977–16987).

Moreover, the double layer capacitance can be correlated to the ECSA using the specific capacitance of the material. We assume the general specific capacitance of 60 $\mu\text{F cm}^{-2}$ of the catalysts (*J. Am. Chem. Soc.* **2013**, *135*, 16977–16987; *Angew. Chem. Int. Ed.* **2014**, *53*, 14433–14437; *J. Am. Chem. Soc.* **2015**, *137*, 4347–4357; *Energy Environ. Sci.* **2015**, *8*, 3022–3029). By calculation and analysis (Detailed calculation method can be found in the **Supplementary Note 2**), the electrochemical active surface area (ECSA) of WS₂ MSLs, 1T'-WS₂ NSs and 2H-WS₂ NSs are 455.0 cm²_{ECSA}, 251.6 cm²_{ECSA}, 180.0 cm²_{ECSA}, respectively.

Referring to the standard of the data diagram in Jaramillo's article, we use the current density as the units on the y-axis instead of the previous " $\Delta J_{0.2V}$ ". Meanwhile, we have added more detailed data analysis in the Supplementary Information (**Supplementary Figure 18–22**).

Figure R24. Nyquist plot obtained by EIS fitted to the Randles cell (inset).

Table R6. Impedance parameters for the equivalent circuit that was shown in **Figure R24**.

Samples	R_s	CPE	R_{ct}
WS ₂ MSLs	0.39	10.8	8.9
1T'-WS ₂ NSs	0.17	15.1	10.5
2H-WS ₂ NSs	0.10	27.3	57.6

Table R7. Impedance parameters for the equivalent circuit that was shown in **Figure R24**.

Samples	EIS— C_{dl} (mF cm ⁻²)	CV— C_{dl} (mF cm ⁻²)
WS ₂ MSLs	27.3	23.8
1T'-WS ₂ NSs	15.1	15.2
2H-WS ₂ NSs	10.8	11.4

We have included these results in the Supplementary Information, in **Supplementary Table 4**, **Supplementary Figure 18–22**, together with discussions. The relevant discussions have been added into the revised manuscript (Page 13, line 1–16).

12. The authors state: “**The much smaller Tafel slopes of WS₂ MSLs (40 mV decade⁻¹) indicated that the kinetics of the electrochemical hydrogen evolution on WS₂ MSLs was much faster than those of the 2H-WS₂ NSs and 1T'-WS₂ NSs (Fig. 3b).**” Tafel slopes DO NOT indicate on kinetics, rather they can indicate on the reaction mechanism under some circumstances (i.e., for a well-defined, dimensionally stable electrode). As noted above, j_0 would be an indicator of kinetics, but this has not been calculated.

Response: Thank you for the constructive comments. The Tafel slope is an important parameter to evaluate the dominant reaction mechanism in the HER process. We agree with you that the Tafel slope don't indicate kinetics, rather it can be used as a guide for the identification of HER mechanism under some circumstances (*ACS Catal.* **2017**, *7*, 7126–7130; *J. Phys. Chem. C* **2019**, *123*, 24007–24012; *Nat. Chem.* **2014**, *6*, 248–253; *Nano Lett.* **2011**, *11*, 4168–4175). We have corrected our conclusion derived from the corrected Tafel slope plots in the revised manuscript (Page 12, lines 7–10).

As shown in **Figure R25b**, according to the reviewer's suggestion, the exchange current density for different studied WS₂ samples was calculated by using extrapolation methods through the Tafel slope plots. We have added the conclusion about exchange current density in the revised manuscript. (Page 12, lines 13–18).

Figure R25. (a) The Tafel slope plots of various catalysts. (b) Exchange current density

calculation. Calculated exchange current for different studied WS₂ samples by using extrapolation methods.

13. The authors state: “**Additionally, compared to 2H-WS₂ and 1T'-WS₂ NSs, the lower charge transfer resistance and rapidly electron transportation capability of the WS₂ MSLs are confirmed by the electrochemical impedance spectroscopy (EIS) measurements (Supplementary Fig. 11).**” However, they do not offer any justification/discussion as to why the EIS measurements indicate this. They do not even fit the spectra or include an equivalent circuit. Given the high porosity of the electrodes in question, interpreting the EIS spectra is not straightforward.

Response: Thanks for the comments and suggestions from the reviewer. Electrochemical impedance spectroscopy (EIS) measurements of the samples were performed using a 100 kHz–0.1 Hz frequency range and an amplitude of 10 mV at 0.25 V (vs. RHE). In the high frequency limit and under non-Faradaic conditions, the electrochemical system is approximated by the modified Randles circuit shown in the inset, where R_s denotes the solution resistance, CPE is a constant-phase element related to the double-layer capacitance, and R_{ct} is the charge-transfer resistance from any residual Faradaic processes. A semicircle in the low-frequency region of the Nyquist plots represents the charge transfer process, with the diameter of the semicircle reflecting the charge-transfer resistance (**Figure R26**). The simulation of the EIS spectra using an equivalent circuit model allowed us to determine the charge transfer resistance, R_{ct} , which is a key parameter for characterizing the catalyst-electrolyte charge transfer process.

As shown in **Table R8**, the fitted R_{ct} values for WS₂ MSLs, 1T'-WS₂ NSs and 2H-WS₂ NSs are 8.9, 10.5, and 57.6 Ω , respectively. Importantly, the WS₂ MSLs exhibits the smallest R_{ct} value among the tested samples, suggesting the superior interfacial charge-transfer kinetics on the surface of WS₂ MSLs for HER catalysis. The R_{ct} values follow an order consistent with the HER performance. (All fitting parameters are listed in **Table R8**).

We attribute this measured small charge transfer resistance (R_{ct}) to its distorted nanobelt structure. The edge-terminated feature can ensure an isotropic electron transport from CFC substrate to WS_2 edges and significantly decrease the resistance for traversed layers (*Nat. Commun.* **2015**, *6*, 7493; *Nano Lett.* **2014**, *14*, 553–558). In addition, the misorientation-induced lattice strain and the reduced interlayer potential barrier in the moiré superlattice (**Figure R27**) can bring beneficial structural and electronic modulations, enabling each WS_2 layer to resemble more closely to a monolayer structure and a better overall conductivity of our WS_2 MSLs (*ACS Energy Lett.* **2019**, *4*, 2830–2835; *Science* **2020**, *370*, 442–445). Moreover, we have added the equivalent circuit model to Nyquist plots in **Supplementary Figure 22**.

Figure R26. Nyquist plots for EIS measurements of WS_2 MSLs, $1T'$ - WS_2 NSs and $2H$ - WS_2 NSs, using the frequency in the range from 100 kHz to 0.1 Hz at 0.25 V (vs. RHE). The inset is the equivalent circuit model that contains the electrolyte resistance (R_s), constant phase element (CPE) and charge-transfer resistance (R_{ct}). Z' is the real impedance and Z'' is the imaginary impedance.

Table R8. Impedance parameters for the equivalent circuit that was shown in **Figure R26**.

Samples	R_s	CPE	R_{ct}
WS_2 MSLs	0.39	10.8	8.9

1T'-WS ₂ NSs	0.17	15.1	10.5
2H-WS ₂ NSs	0.10	27.3	57.6

Figure R27. Total potentials for the superlattice with $\theta = 0$ and 13.2° .

We have included these results in the Supplementary Information, **Supplementary Figure 22, 23, Supplementary Table 4**, together with discussions. The relevant discussions have been added into the revised manuscript (Page 13, line 16–22, page 14, line 1–5).

14. The authors state: “**The electrochemical double-layer capacitances (C_{dl}) were calculated to contrast the electrochemical surface area (ECSA) of 2H-WS₂ NSs, 1T'-WS₂ NSs and WS₂ MSLs (Fig. 3d). The C_{dl} of WS₂ MSLs (33.7 mF cm^{-2}) was much higher than that of 1T'-WS₂ NSs (21.2 mF cm^{-2}) and 2H-WS₂ NSs (7.2 mF cm^{-2}), indicating that the WS₂ MSLs possessed more fully exposed active sites for electrochemical hydrogen evolution.**” How were these values normalised? Geometric area? In principle, if the specific capacitance (C_{specific} in F cm^{-2}) of these materials was known, then the exposed surface area (or ECSA) could be estimated from these values (i.e., $A_{\text{ECSA}} = C_{dl}/C_{\text{specific}}$). The ECSA however does not necessarily indicate on the number of exposed “active sites”, as all sites, regardless of ‘activity’ (e.g., the basal and edge planes of 2H-WS₂ plus the underlying carbon support) contributes to the non-

faradaic current, whereas only certain sites (e.g., the edge plane of 2H-WS₂) may dominate the HER catalysis. How was the carbon support corrected for when calculating the ECSA?

Response: Thanks for the comments and suggestions from the reviewer. As shown in **Figure R20-23**, the specific capacitance values of WS₂ MSLs, 1T'-WS₂ NSs and 2H-WS₂ NSs are 23.8 mF cm⁻², 15.2 mF cm⁻², 11.4 mF cm⁻² respectively.

In order to confirm the determination of the electrochemically active surface area (ECSA) of the catalysts, the following equation (*J. Am. Chem. Soc.* **2013**, *135*, 16977–16987; *Angew. Chem. Int. Ed.* **2014**, *53*, 14433–14437; *J. Am. Chem. Soc.* **2015**, *137*, 4347–4357; *Energy Environ. Sci.* **2015**, *8*, 3022–3029) could be utilized below:

$$A_{ECSA} = \frac{C_{dl}}{C_{specific}}$$

where C_{dl} is double-layer capacitance, C_{specific} is specific capacitance.

In general, the ECSA estimates to be accurate within about an order of magnitude, and emphasize that the values should be considered only as an approximate guide for comparing electroactive surface area. The specific capacitance is converted into an electrochemical active surface area (ECSA) using the specific capacitance value for a flat standard with 1 cm² of real surface area. The specific capacitance for a flat surface is generally found to be in the range of 20–60 μF cm⁻². In the following calculations of ECSA, we assume the value of 60 μF cm⁻² as the specific capacitance of the catalysts in this work (*J. Am. Chem. Soc.* **2015**, *137*, 4347–4357; *Energy Environ. Sci.* **2015**, *8*, 3022–3029; *ACS Catal.* **2014**, *4*, 3957–3971; *Nat. Commun.* **2019**, *10*, 2281). Based on the above equation analysis, the detailed calculation process is as follows:

Calculated electrochemical active surface area:

I: WS₂ MSLs II: 1T'-WS₂ NSs III: 2H-WS₂ NSs

$$A_{ECSA}^I = \frac{23.8 \text{ mF cm}^{-2}}{60 \text{ } \mu\text{m cm}^{-2} \text{ per cm}_{ECSA}^2} = 396.6 \text{ cm}_{ECSA}^2$$

$$A_{ECSA}^{II} = \frac{15.2 \text{ mF cm}^{-2}}{60 \mu\text{m cm}^{-2} \text{ per cm}_{ECSA}^2} = 253.3 \text{ cm}_{ECSA}^2$$

$$A_{ECSA}^{III} = \frac{11.4 \text{ mF cm}^{-2}}{60 \mu\text{m cm}^{-2} \text{ per cm}_{ECSA}^2} = 190.0 \text{ cm}_{ECSA}^2$$

The electrochemical active surface area (ECSA) of WS₂ MSLs (396.6 cm²) is much higher than those of 2H-WS₂ NSs (253.3 cm²) and 1T'-WS₂ NSs (190.0 cm²), indicating that the WS₂ MSLs possesses more abundant active sites.

It is well known that bare carbon fiber cloth (CFC) preferentially increases the conductivity and dispersity of the samples so that higher current densities and lower overpotentials for electrocatalysis will be exhibited. But the bare CFC we used, Phychemi (HK) Company Limited-W0S1010, can hardly be polarized for HER (seeing the curve in **Figure R28**). Therefore, the intrinsic performances of the bare carbon for HER can be neglected. Moreover, the same electrochemical capacitance test methods (see the part of Electrochemical Measurements in Supporting Information) were used to determine the double-layer capacitance (C_{dl}) of bare CFC (**Figure R29**). The C_{dl} values of the different samples for HER were shown in **Table R9**. The C_{dl} of all WS₂ samples are more than two orders of magnitude greater than that of bare CFC. Therefore, the bare CFC exhibits negligible effect on the electrochemical surface area measurement of electrode materials. The background capacitance of the bare CFC electrode was subtracted from the obtained double layer capacitance to compensate for the low substrate coverage (*Nat. Commun.* **2019**, *10*, 2650).

Figure R28. The polarization curve of bare CFC.

Figure R29. Double-layer capacitance measurements for determining electrochemically-active surface area for the substrate material (bare CFC) from voltammetry in Ar-bubbled 0.5 M H₂SO₄. (a) Cyclic voltammograms were measured in a non-Faradaic region of the voltammogram at the following scan rate: 10, 20, 40, 60, 80, 100, 120, and 140 mV s⁻¹. The working electrode was held at each potential vertex for 10 s before the beginning the next sweep. All current is assumed to be due to capacitive charging. (b) The cathodic (○) and anodic (□) charging currents measured at 0.275 V vs. RHE plotted as a function of scan rate.

Table R9. The C_{dl} values of the different samples for HER.

Samples	2H-WS ₂ NSs	1T'-WS ₂ NSs	WS ₂ MSLs	Bare CFC
C _{dl} (mF cm ⁻²)	11.4	15.2	23.8	0.0876

We have included these results in the Supplementary Information, in **Supplementary Table 3, Supplementary Figure 18**, together with discussions. We have included the details about ECSA calculations in the **Supplementary Note 1**. The relevant discussions have been added into the revised manuscript (Page 13, line 1–16).

15. The authors use the terms “**superhydrophilic**” and “**superaerophobic**” throughout. It would be useful if they provide a definition of these terms for the reader (i.e., what distinguishes hydrophilic from superhydrophilic?).

Response: Thanks for reviewer’s kind suggestion. According to the comments of reviewers, in the revised manuscript, we try to clarify some basic concepts and facilitate readers to better understand the unique wetting phenomenon on the electrode material surface. We have added some terms interpretation in the revised manuscript, such as “hydrophilic”, “hydrophobic” and “superhydrophilic” (Page 16, line 13–16). In addition, we have added the **Supplementary Figure 25 (Figure R30)** to help readers better understand these terms, showing a detailed schematic diagram and corresponding textual annotations.

Figure R30. Schematic of the different wetting states that are possible on catalytic surfaces.

16. In Figure 5, it is not clear to me what the numbers “1,2,3,4,5,6” refer to or what the arrows are indicating. It is also not clear why a 3D plot is necessary here? The pictures of the various active sites are also too small to see clearly, so overall it is very difficult to determine anything meaningful from this figure.

Response: Thanks for reviewer's kind suggestion. We have changed the constitution of Figure 5 to help readers comprehend the meaningful content better. In order to make Figure 5 clear, the sufficiently clear model diagrams (side view) of various adsorption structures were shown in Figure 5a in the revised manuscript, and more details about these adsorption structures were shown in **supplementary Figure 31–36**. Moreover, the changes in the ΔG_H values of various active sites corresponding to the different atomic structure models (**Figure 5a**) were shown in **Figure 5b**. Meanwhile, we have added some illustrations and textual annotations to eliminate readers' confusions about **Figure 5**. We added the **Figure R31** as a new **Figure 5** in the revised manuscript (Page 18).

Figure R31. (a) Adsorption structures of H* at the W-edge site of normal WS₂ (Model-1), W-edge site-1 of rotated WS₂ (Model-2), W-edge site-2 of rotated WS₂ (Model-3), S-edge site of normal WS₂ (Model-4), S-edge site-1 of rotated WS₂ (Model-5) and S-edge site-2 of rotated WS₂ (Model-6) (side view). Yellow, cyan, and white balls represent S, W, and adsorbed H atoms. (b) The changes (indicated by the arrows) of ΔG_H values of various active sites which corresponded to the different atomic structure models in Figure 5a.

17. The authors state “We ascribe the activity enhancement to a combination of electronic, geometric, superaerophobic and superhydrophilic effects.” Again, as stated above, in electrochemistry “activity” refers to electron-transfer kinetics. This statement is misleading, as the aforementioned “superaerophobic and superhydrophilic effects” influence the mass transfer of bubbles, rather than enhancing electron-transfer kinetics.

Response: Thanks for reviewer’s kind suggestion. The two primary categories of activity measurements are “total electrode” activity (i.e., geometric electrode area-normalized measurements) and “intrinsic” activity (i.e., per-site turnover frequency, TOF). We have carefully polished this statement to “We ascribe the total electrode activity enhancement to a combination of electronic, geometric, superaerophobic and superhydrophilic effects.” We have made the corresponding changes in the revised manuscript (Page 20, line 22; page 21, line 1–2).

18. The authors state “ ΔG_H is insensitive to the MSLs” which is counter to the argument that they presented in Figure 5 (e.g., “**Computational predictions for the MSLs effect on the HER activity indicated that the active sites of W-edge and S-edge of twisted bilayers WS₂ have much more appropriate ΔG_H compared with normal bilayers WS₂ in Fig. 5.**”)

Response: Thanks for reviewer’s kind suggestion. We have revised some confusing statements in the revised manuscript. (Page 21, line 4–5)

19. (1) The “Electrochemical Measurements.” Section of the SI is not sufficiently detailed to enable the reader to repeat the measurements. For example, in the “supplementary methods section” the authors state **“The full description of linear sweep voltammetry (LSV) and cyclic voltammetry (CV) tests have been shown in supplementary methods.”** Also **“The value of electrochemical double layer capacitance (C_{dl}) Electrochemical active surface areas (ECSA) was calculated by measuring CV curves of samples.”** is very ambiguous.

(2) Further **“Nyquist plots of three samples were measured in the frequency range from 100 Hz to 0.1 kHz at an open circuit potential of -350 mV.”** According to Figure 3a of the main text, -350 mV is well beyond the onset potential of the HER on all considered electrodes. Therefore, it is simply impossible that -350 mV could correspond to the “open circuit potential” of the catalysts.

Response (1): Thanks for reviewer’s kind suggestion. We have added sufficient details to enable the reader to repeat the measurements in the Supplementary Information (see the part of Electrochemical Measurements, page 4–6, marked by red). For your convenience, the related corrections are as follows:

“All the electrochemical experiments were carried out using a conventional three-electrode system on an Electrochemical Workstation (CS310, Wuhan Kesite Instrument Co., Ltd.). All electrochemical performance tests of samples were carried out on carbon fiber cloth (CFC) (Phychemi (HK) Company Limited-W0S1010). A typical three-electrode configuration was used to investigate all samples HER performance with an Ag/AgCl electrode and a graphite rod as the reference and counter electrodes, respectively. All the electrochemical measurements were conducted in Ar-bubbled 0.5 M H_2SO_4 electrolyte at room temperature. All the potentials reported in this work were normalized against that of the reversible hydrogen electrode (RHE).

Before the electrochemical test, the fresh as-prepared 1T'- WS_2 NSs product and 2H- WS_2 NSs was added into a 100 mL Erlenmeyer flask containing 3 mL thioglycolic acid and 50 mL ethanol, and vigorously stirred for 12 h under N_2 atmosphere to partially

removing the surfactant molecules. After that, the acid-treated 1T'-WS₂ NSs were separated from the solution by centrifugation (8,500 rpm, 10 min), washed twice with ethanol. A catalyst dispersion was prepared by mixing 2.5 mg of the active material, namely WS₂ MSLs, 1T'-WS₂ NSs, 2H-WS₂ NSs hybrid in a solution containing 10 μ L of Nafion (5 wt %) aqueous solution, 400 μ L D.I. water and 100 μ L absolute ethanol, followed by ultrasonication for 45 min. Then, 40 μ L of the catalyst dispersion was dropped onto the surface of cleaned CFC (1 cm²) (equivalent to 0.196 mg cm⁻²) and dried overnight naturally.

The LSV polarization curves were obtained by sweeping the potential from -0.35 to 0.10 V vs. RHE at room temperature and the scan rate was 10 mV s⁻¹. The sweep rate of 10 mV s⁻¹ we used is slow enough to build a steady state electrode and thus the resulting polarization curve is reasonably to be used for kinetic analysis (*Nat. Commun.* **2015**, *6*, 5982). All polarization curves were obtained with iR compensation. The polarization curves were replotted as overpotential (η) versus log current ($\log J$) to get Tafel plots for assessing the HER kinetics of investigated catalysts. By fitting the linear portion of the Tafel plots to the Tafel equation ($\eta = b \log (J) + a$), the Tafel slope (b) can be obtained. From the intercept of the linear region of Tafel plot, the exchange current density (j_0 , the intrinsic electron transfer rate between the electrode and the electrolyte) is obtained for the different WS₂ samples.

Cyclic voltammetry (CV) was conducted to evaluate the electrochemical double layer capacitance (C_{dl}) of the materials at non-faradaic potentials as the means of estimating the corresponding electrochemical active surface areas. For CV measurements, a series of CV curves were performed at various scan rates (10, 20, 40, 60, 80, 100, 120, 140 mV s⁻¹) in 0.25–0.30 V vs. RHE region. The cathodic (\ominus) and anodic ($\omin�$) charging currents measured at 0.275 V (vs. RHE) plotted as a function of scan rate. The determined double-layer capacitance of the system is taken as the average of the absolute value of the slope of the linear fits to the data. The values of C_{dl} were used to estimate the electrochemically active surface area (ECSA) (Calculation details are provided in the **Supplementary Note 2**).

The chronoamperometry test was performed to evaluate the catalyst's durability during catalysis. WS₂ MSLs, 1T'-WS₂ and 2H-WS₂ NSs-coated carbon fiber cloth (1 cm², catalyst loading 280 μg) were used as working electrodes to collect chronoamperometry data at the applied potential of -0.20 V vs. RHE.

The uncompensated resistance was measured by electrochemical impedance spectroscopy (EIS). The EIS measurements were performed in the same configuration at 0.25 V (vs. RHE) from 100 kHz to 0.1 Hz. The electrolyte resistance (R_s) was measured using EIS measurements and used for iR compensation by the equation of $E_{iR-corrected} = E_{original} - I \times R_s$ (Nat. Commun. 2020, 11, 3315; Nat Commun. 2018, 9, 1425)."

Response (2): Thanks for reviewer's kind suggestion. We remeasured the EIS experiments at 0.25 V (vs. RHE) over a frequency range from 100 kHz to 0.1 Hz at the room temperature. Electrochemical impedance spectroscopy with the fitted circuit model shows significantly decreased charge-transfer resistances (R_{ct}) for the WS₂ MSLs (8.9 Ω) samples, as compared to those of 1T'-WS₂ NSs (10.5 Ω) and 2H-WS₂ NSs (57.6 Ω), indicating the lower charge transfer resistance and facilitated electron transportation capability of the WS₂ MSLs (see the data in **Figure R32** and **Table R10**).

Figure R32. Nyquist plots for EIS measurements of WS₂ MSLs, 1T'-WS₂ NSs and 2H-WS₂ NSs, using the frequency in the range from 100 kHz to 0.1 Hz at 0.25 V (vs. RHE). The inset is the equivalent circuit model that contains the electrolyte resistance (R_s), constant phase element (CPE) and charge-transfer resistance (R_{ct}). Z' is the real

impedance and Z'' is the imaginary impedance.

Table R10. Impedance parameters for the equivalent circuit that was shown in **Figure R32**.

Samples	R_s	CPE	R_{ct}
WS ₂ MSLs	0.39	10.8	8.9
1T'-WS ₂ NSs	0.17	15.1	10.5
2H-WS ₂ NSs	0.10	27.3	57.6

We have included these results in the Supplementary Information, in **Supplementary Table 4**, **Supplementary Figure 22**, together with discussions. The relevant discussions have been added into the revised manuscript (Page 13, line 16–20).

Reply to Referee 3 and revisions made accordingly:

In this manuscript, the authors reported the synthesis of large-scale WS₂ Moiré superlattices (MSLs) through a one-pot hydrothermal approach, and demonstrated their catalytic hydrogen production performance. However, the evidence of Moiré superlattices is not sufficient and doesn't support the main theme of this article. Thus, the main claim in this work is weak. Upon a careful examination, I cannot recommend its publication in Nature Communications.

Response: We thank the referee for carefully reviewing our manuscript and the raised valuable comments. For the structural analysis, we have provided more detailed and convincing evidences of Moiré superlattices to support the main theme of our article. Our point-by-point response is presented as follows.

1. HRTEM images in Figure 2a and 2b seems have been applied too much filters during the data recording and processing, and the images didn't show a clear Moiré period even in a small region.

Response: Thanks for the comments and suggestions from the reviewer. Moiré patterns can arise under two conditions, either when the two lattices have slightly different parameters or when identical lattices are twisted at an angle θ with respect to each other. HRTEM is widely used to characterize moiré patterns in 2D materials. In order to demonstrate that the synthesized WS₂ was moiré superlattices (MSL) material, we provide HRTEM images and the simulated HRTEM image of WS₂ as the sufficient evidence to support that it is the MSL material. We collected low-magnified TEM images and estimated the moiré superlattices as shown in **Figure R33**. Low-magnified TEM image of WS₂ MSL in **Figure R33a** exhibits a well-arranged hexagonal lattice structure which is attributed to the twist of bilayer WS₂ with a twisted angle. As can be seen, moiré superlattices are found throughout the measured region. The corresponding FFT patterns contain double sets of 6-fold symmetry diffraction spots. According to the measurement of the splitting spots in the FFT patterns, the misorientation angle of

$\sim 13.8^\circ$ could be calculated from the fast Fourier transformed (FFT) images as shown in **Figure R33b**. Herein, the twisted angle θ also could be obtained via the formula: $\theta = 2\arcsin a/\lambda$, where $a = 0.322$ nm is the lattice constant of WS_2 and $\lambda \approx 1.34$ nm is the moiré wavelength depicted in **Figure R33c** (*Nature* **2018**, 556, 80–84). The significant honeycomb-structured Moiré pattern in **Figure R33c** is consistent with the simulated HRTEM images of WS_2 MSL (**Figure R33d**).

Figure R33. (a) Low-magnified TEM image of WS_2 MSLs, (b) The corresponding FFT pattern of (a), (c) HRTEM image of WS_2 MSLs, (d) Simulated HRTEM image of WS_2 MSLs.

To address the reviewer's concern, we replaced some typical images with regular hexagonal MSLs domains in **Figure 2a, b** as sufficient evidence to support that it is the MSL material (page 7 in the revised manuscript). The relevant discussions have been

added into the revised manuscript (Page 7, 2–4, page 8, line 5–11).

2. From the XRD results in Fig. S5, we can see the 1T'-WS₂ NSs sample has a low crystallinity, to make a comparison, the standard PDF card should be presented in the same image.

Response: Thanks for the referee's valuable comments and suggestion. For comparison, the standard PDF card (PDF#08-0237) was presented in XRD pattern (**Figure R34**). We added the **Figure R34** as a new **Supplementary Figure 8** in Supplementary Information.

Figure R34. XRD pattern of 1T' WS₂ NSs. Using PDF#08-0237 for reference.

3. The EIS measurements carried out at a large overpotential (-350 mV), which is not appropriate. The EIS measurements are better carry out at a small catalytic current region.

Response: Thanks for reviewer's kind suggestion. We remeasured the EIS experiments at 0.25 V (vs. RHE) over a frequency range from 100 kHz to 0.1 Hz at the room temperature. Electrochemical impedance spectroscopy with the fitted circuit model shows significantly decreased charge-transfer resistances (R_{ct}) for the WS₂ MSLs (8.9 Ω) samples, as compared to those of 1T'-WS₂ NSs (10.5 Ω) and 2H-WS₂ NSs (57.6 Ω), indicating the lower charge transfer resistance and facilitated electron transportation

capability of the WS₂ MSLs (see the data in **Figure R35** and **Table R11**).

Figure R35. Nyquist plots for EIS measurements of WS₂ MSLs, 1T'-WS₂ NSs and 2H-WS₂ NSs, using the frequency in the range from 100 kHz to 0.1 Hz at 0.25 V (vs. RHE). The inset is the equivalent circuit model that contains the electrolyte resistance (R_s), constant phase element (CPE) and charge-transfer resistance (R_{ct}). Z' is the real impedance and Z'' is the imaginary impedance.

Table R11. Impedance parameters for the equivalent circuit that was shown in **Figure R35**.

Samples	R_s	CPE	R_{ct}
WS ₂ MSLs	0.39	10.8	8.9
1T'-WS ₂ NSs	0.17	15.1	10.5
2H-WS ₂ NSs	0.10	27.3	57.6

We have included these results in the Supplementary Information, in **Supplementary Table 4**, **Supplementary Figure 22**, together with discussions. The relevant discussions have been added into the revised manuscript (Page 13, line 16–20).

4) In line2, page 7. The “S-Mo-S” should be S-W-S.

Response: We appreciate the reviewer very much for the helpful suggestion. We have changed the “S-Mo-S” to “S-W-S”, which has been highlighted in the revised

manuscript (page 6, line 19).

Considering the novelty, scientific importance and potential applications, we wish the reviewer can share our enthusiasms that this work indeed represents one of the most important findings in the current development of Moiré superlattices catalysts enhancing HER mechanism and it deserves to be published in *Nature Communications*.

REVIEWER COMMENTS

Reviewer #1 (Remarks to the Author):

The authors have answered all my questions and I accept it for publication.

Reviewer #2 (Remarks to the Author):

I have read the revised manuscript and responses to my original comments in detail and agree that there is a significant improvement in this version. However, I still have some queries about the electrochemical dataset. Please consider the following:

1. Lines 68 – 69: “The low Tafel slope of 40 mV decade⁻¹ for HER indicates the fast reaction kinetics.” Please clarify this comment, as Tafel slope does not indicate on charge-transfer kinetics. I already addressed this in my previous comments (Reviewer 2, comment 12)

2. Figure 1c: No units have been included on the color bar. Is this a relative scale? If so, state it in the caption.

3. Lines 204 – 208: Exchange current density is only meaningful if normalised to the ECSA (see Reviewer 2, comment 9). I suggest moving this discussion to after the ECSA has been calculated (perhaps where the TOF is discussed?) and re-normalising j_0 with the ECSA.

4. In response to Reviewer 2, comment 2: In my view, the revised text is still ambiguous. Ultimately, it is the decision of the authors, but my recommendation would be “ascribed to a closer to thermoneutral hydrogen adsorption free energy value (i.e., $\Delta G_H \rightarrow 0$) of twisted bilayers active sites”

5. In response to Reviewer 2, comment 6: In Figure R11, “at a static overpotential of 0.2 V vs. RHE”. Potential is measured versus a reference (RHE) whereas overpotential is measured relative to the equilibrium potential, by definition. This statement should either read: “at a static potential of -0.2 V vs. RHE” or “at a static overpotential of 0.2 V”

6. In response to Reviewer 2, comment 13: This remains a concern in the revisions. In Figure R26 was truly taken in a “non-faradaic” region, that how can the authors justify the use of a Randles circuit, which is used to describe a Faradaic process? Strictly speaking, if the electrode/electrolyte behaves as an “ideal” electrochemical capacitor (Reviewer 2, comment 11), then there should be no charge transfer (Faradaic) process taking place. What is the physical origin of RCT? In addition, given that this RCT value cannot be related to the HER, as it is thermodynamically impossible at 0.25 V vs. RHE, then why should RCT be related to HER activity? In my view, there is no physicochemical reason why the RCT measured for a charge-transfer process at 0.25 V vs. RHE should reflect the kinetics of the HER, which is a totally different process that takes place in a different potential range!

Point-by-point response to the referees' comments

We thank the referees for their valuable comments and positive endorsement to our manuscript. We have carefully considered the referees' comments and revised the manuscript accordingly. Our responses and corresponding revisions are as follows:

Reply to Referee 1:

The authors have answered all my questions and I accept it for publication.

Response: We are very grateful to your encouraging and positive comments and really appreciate your agreement of acceptance with this revised manuscript.

Reply to Referee 2 and revisions made accordingly:

I have read the revised manuscript and responses to my original comments in detail and agree that there is a significant improvement in this version. However, I still have some queries about the electrochemical dataset. Please consider the following:

Response: Many thanks for your positive comments on our manuscript. We have revised our manuscript accordingly.

1. Lines 68–69: “The low Tafel slope of 40 mV decade⁻¹ for HER indicates the fast reaction kinetics.” Please clarify this comment, as Tafel slope does not indicate on charge-transfer kinetics. I already addressed this in my previous comments (Reviewer 2, comment 12)

Response: Thanks for the comments from the reviewer. We have deleted the inappropriate statement — “The low Tafel slope of 40 mV decade⁻¹ for HER indicates the fast reaction kinetics.” In addition, we have changed “The as-synthesized WS₂ MSLs electrocatalysts display the best activity as manifested by the lowest overpotential of 60 mV versus reversible hydrogen electrode (RHE) at 10 mA cm⁻². The low Tafel slope of 40 mV decade⁻¹ for HER indicates the fast reaction kinetics.” to “The as-synthesized WS₂ MSLs electrocatalysts display an overpotential of 60 mV at a current density of 10 mA cm⁻² and a Tafel slope of 40 mV dec⁻¹” in the revised manuscript (Page 4, line 9–11).

2. Figure 1c: No units have been included on the color bar. Is this a relative scale? If so, state it in the caption.

Response: Thanks for the comments and suggestions from the reviewer. The color bar shows the relative scale of the strain distribution. We have added the statement about the color bar in the caption (Page 6, line 2).

3. Lines 204–208: Exchange current density is only meaningful if normalised to the

ECSA (see Reviewer 2, comment 9). I suggest moving this discussion to after the ECSA has been calculated (perhaps where the TOF is discussed?) and re-normalising j_0 with the ECSA.

Response: Thanks for the comments and suggestions from the reviewer. As shown in **Figure R1**, the exchange current density was normalized to electrochemically active surface area (ECSA). Moreover, we moved the discussion about exchange current density to after the ECSA calculation and before the TOF calculation in the revised manuscript (Page 14, line 1–5). We added the **Figure R1** as a new **Supplementary Figure 24** in Supplementary Information.

Figure R1. (a) Tafel curves of different catalysts. (b) Comparison of the ECSA and the ECSA-normalized j_0 of (I) WS₂ MSLs, (II) 1T'-WS₂ NSs, (III) 2H-WS₂ NSs.

4. In response to Reviewer 2, comment 2: In my view, the revised text is still ambiguous. Ultimately, it is the decision of the authors, but my recommendation would be “ascribed to a closer to thermoneutral hydrogen adsorption free energy value (i.e., $\Delta G_{\text{H}} \rightarrow 0$) of twisted bilayers active sites”

Response: Thanks for the comments and suggestions from the reviewer. After careful consideration, we believe that the statement suggested by reviewer is clearer and more

accurate. We have used “ascribed to a closer to thermoneutral hydrogen adsorption free energy value of twisted bilayers active sites” instead of “ascribed to appropriate ΔG_H of twisted bilayers WS_2 active sites” in the revised manuscript (Page 2, line 11–13).

5. In response to Reviewer 2, comment 6: In Figure R11, “at a static overpotential of 0.2 V vs. RHE”. Potential is measured versus a reference (RHE) whereas overpotential is measured relative to the equilibrium potential, by definition. This statement should either read: “at a static potential of -0.2 V vs. RHE” or “at a static overpotential of 0.2 V”

Response: Thanks for the comments and suggestions from the reviewer. We have used “at a static potential of -0.2 V vs. RHE.” instead of “at a static overpotential of 0.2 V vs. RHE.” in the revised manuscript (Page 11, line 5–6).

6. In response to Reviewer 2, comment 13: This remains a concern in the revisions. In Figure R26 was truly taken in a “non-faradaic” region, that how can the authors justify the use of a Randles circuit, which is used to describe a Faradaic process? Strictly speaking, if the electrode/electrolyte behaves as an “ideal” electrochemical capacitor (Reviewer 2, comment 11), then there should be no charge transfer (Faradaic) process taking place. What is the physical origin of R_{CT} ? In addition, given that this R_{CT} value cannot be related to the HER, as it is thermodynamically impossible at 0.25 V vs. RHE, then why should R_{CT} be related to HER activity? In my view, there is no physicochemical reason why the R_{CT} measured for a charge-transfer process at 0.25 V vs. RHE should reflect the kinetics of the HER, which is a totally different process that takes place in a different potential range!

Response: Thanks for the comments and suggestions from the reviewer. We agree with you that the R_{ct} value fitted in the non-faraday process is not be related to the HER, let alone the dynamics of HER.

To illustrate the improved electron transport in HER, electrochemical impedance spectroscopy (EIS) was carried out at a potential of -50 mV (vs. RHE) for all WS_2

samples, as shown in the Nyquist plots in **Figure R2**. The EIS measurements were performed in 0.5 M H₂SO₄ solution from 100 kHz to 0.1 Hz. The Warburg component of the Nyquist plots is not observed at frequencies as low as 0.1 Hz. This indicates that there was a sufficient supply of H⁺ at the surface and mass transport impedance can be neglected¹. The EIS data are characterized by two overlapping semicircles on the complex plane plot; one at high frequencies with small diameter and one at low frequencies with larger diameter. Several models have been developed to explain the two semi-circle EIS response of HER on various electrodes¹.

We adopt the two time constant parallel model proposed by Armstrong and Henderson² (the equivalent circuit is displayed in **Figure R2b**). We added an inductor to account for the observed high frequency inductance from, e.g. the connection wires. The use of constant phase elements instead of capacitors is required to account for the slight depression of the semi-circles caused by inhomogeneities at the atomic scale³⁻⁵. The constant phase element impedance is described by:

$$Z_{\text{CPE}} = \frac{1}{Q(i\omega)^\alpha}$$

where $i = \sqrt{-1}$, ω is the angular frequency of the AC voltage, $0 < \alpha < 1$, and Q is the frequency-independent parameter. The case $\alpha = 1$ recovers a perfect capacitor.

The capacitance associated with the CPS can be evaluated as⁶:

$$C_i = \left(\frac{Q_i}{(R_S^{-1} + R_i^{-1})^{1-\alpha}} \right)^{\frac{1}{\alpha}}$$

This model is characterized by one high frequency time constant (τ_1 , CPE₁-R₁) and one low frequency time constant (τ_2 , CPE₂-R₂). The experimental impedance data were fitted to the equivalent circuit model by ZView software. The EIS data at a potential of -50 mV vs. RHE together with the corresponding fitted curve are displayed in **Figure R2a** and the equivalent circuit model is seen to the EIS data very well. In addition, it is noted that equivalent circuits drawn for same electrochemical impedance spectroscopy

are not unique.

In the two time constant parallel model the R_s resistance element is attributed to the uncompensated solution resistance, the high frequency time constant (τ_1 , CPE₁-R₁) is related to the Faradaic resistance for the charge transfer process, R_{ct} , and double layer capacitance, C_{dl} , while the low frequency time constant (τ_2 , CPE₂-R₂) is related to hydrogen adsorption, R_p and C_p .

The fitted R_{ct} values for WS₂ MSLs, 1T'-WS₂ NSs and 2H-WS₂ NSs are 1.6, 3.4, and 11.2 Ω , respectively (**Figure R2**). Importantly, the WS₂ MSLs exhibits the smallest R_{ct} value among the tested samples, suggesting the superior interfacial charge-transfer kinetics on the surface of WS₂ MSLs for HER catalysis.

We added the **Figure R2** as a new **Supplementary Figure 24** in Supplementary Information. We have corrected our conclusion derived from EIS in the revised manuscript (Page 13, lines 11–14).

Figure R2. (a) Nyquist plots of impedance data for different WS₂ samples. (b) The equivalent circuit used to fit the data.

Reference

1. Lasia, A. in *Modern Aspects of Electrochemistry*, Vol. 32. (eds. B.E. Conway, J.O.M. Bockris & R. White) 143-248 (Springer US, 2002).
2. Armstrong, R. D. & Henderson, M. Impedance plane display of a reaction with an adsorbed intermediate. *J. Electroanal. Chem. Interfacial Electrochem.* **39**, 81–90

- (1972).
3. Kerner, Z. & Pajkossy, T. On the origin of capacitance dispersion of rough electrodes. *Electrochim. Acta* **46**, 207–211 (2000).
 4. Navarro-Flores, E., Chong, Z. & Omanovic, S. Characterization of Ni, NiMo, NiW and NiFe electroactive coatings as electrocatalysts for hydrogen evolution in an acidic medium. *J. Mol. Catal. A: Chem.* **226**, 179–197 (2005).
 5. Kibsgaard J, Jaramillo TF, Besenbacher F. Building an appropriate active-site motif into a hydrogen-evolution catalyst with thiomolybdate $[\text{Mo}_3\text{S}_{13}]^{2-}$ clusters. *Nat. Chem.* **6**, 248–253 (2014).
 6. Brug, G. J., Vandeneeden, A. L. G., Sluytersrehabach, M. & Sluyters, J. H. The analysis of electrode impedances complicated by the presence of a constant phase element. *J. Electroanal. Chem.* **176**, 275–295 (1984).

REVIEWERS' COMMENTS

Reviewer #2 (Remarks to the Author):

I have reviewed the manuscript and am satisfied with the revisions. I am happy to see it published in Nature Communications.